# Fracture toughness of mixed-mode anticracks in highly porous materials

Valentin Adam[1,2], Bastian Bergfeld [2], Philipp Weißgraeber [3], Alec van Herwijnen [2] & Philipp L. Rosendahl [1] ✉

When porous materials are subjected to compressive loads, localized failure chains, commonly termed anticracks, can occur and cause large-scale structural failure. Similar to tensile and shear cracks, the resistance to anticrack growth is governed by fracture toughness. Yet, nothing is known about the mixed-mode fracture toughness for highly porous materials subjected to shear and compression. We present fracture mechanical field experiments tailored for weak layers in a natural snowpack. Using a mechanical model for interpretation, we calculate the fracture toughness for anticrack growth for the full range of mode interactions, from pure shear to pure collapse. The measurements show that fracture toughness values are significantly larger in shear than in collapse, and suggest a power-law interaction between the anticrack propagation modes. Our results offer insights into the fracture characteristics of anticracks in highly porous materials and provide important benchmarks for computational modeling.

Over the past century, fracture mechanics has profoundly impacted material science and engineering, providing a crucial framework to understand and predict material failure. The complexities of fracture mechanics become particularly intriguing when applied to porous materials, where unique challenges arise due to the occurrence of compressive fractures, so-called anticracks, alongside traditional tensile and shear fractures described in classical fracture mechanics. Despite the widespread existence of porous materials in both natural environments and engineering applications, their fracture mechanics, notably the formation of anticracks, has thus far received limited attention.

Porous materials, widely employed in engineering applications from fluid machinery to aerospace[1], reveal a notable vulnerability to compression, resulting in the formation of anticracks. This susceptibility extends beyond foams[2,3] and honeycombs[4] to encompass various materials, including cellular structures[5]. Similar observations can be made in compacting pharmaceutical pills[6], crushing cereal packs[7], the failure mechanisms of bones[8,9], and even in geotechnical materials like rocks[10–12]. Moreover, this phenomenon offers a comprehensive explanation for natural hazards, including deep-focus earthquakes[13],

landslides[14], failure of embankments[15], ground settlement[16], and firn quakes on large glacier sheets[17]. In these instances, specific regions or planes with more fragile properties than the surrounding material, commonly referred to as weak layers, are prone to collapse. A striking example of anticrack propagation in weak layers is the occurrence of slab avalanches[18,19], which result from the collapse of porous weak layers in a stratified snowpack, and constitute the focus of this study.

The key material property for predicting crack extension is fracture toughness, which represents the critical energy release rate necessary for a crack to propagate[20]. Cracks may be driven by opening or closing (mode I), shearing (mode II), or tearing (mode III) deformations, each associated with a certain fracture toughness[21]. In the case of combined normal, in-plane shear, or out-of-plane shear loading (mixed mode), the interaction law between the pure modes and their respective fracture toughnesses must be identified[22]. These interaction laws are available for many natural and engineering materials such as metals[23], rock[24], fiber-reinforced polymers[25], bones[26], or meta materials[27], offering insights into the intrinsic properties and structural integrity. However, for anticracks with closing mode I, no interaction law exists for any material.

[1]Institute of Structural Mechanics and Design, Department of Civil and Environmental Engineering, Technical University of Darmstadt, Franziska-Braun-Str. 3, 64285 Darmstadt, Germany. [2]WSL Institute for Snow and Avalanche Research SLF, Flüelastr. 11, 7260 Davos, Switzerland. [3]Chair of Lightweight Design, Faculty of Mechanical Engineering and Marine Technology, University of Rostock, Albert-Einstein-Straße 2, 18059 Rostock, Germany. ✉e-mail: rosendahl@ismd.tu-darmstadt.de

Experimental measurements of fracture toughness for anticracks in brittle materials are challenging. Aside from notched samples under compressive loading[3,5], the propagation saw test (PST) is employed, in particular for weak layers in snow[28–30]. In the PST, an artificial crack is cut into the weak layer of an isolated snow block until the overhanging cantilever releases the critical energy required for crack growth. However, as anticrack propagation inherently involves mode II, both in flat terrain with a horizontal bending component[31] and more prominently on inclined slopes[18,32], it should be recognized as a mixed-mode process. A comprehensive understanding of anticrack fracture behavior across the entire interaction regime, from pure collapse (mode I) to pure shear (mode II or III), is currently lacking due to the absence of a suitable experimental setup.

While previous studies have focused only on experimental procedures for mode I loading[3], we adopt a multidisciplinary strategy, combining fracture experiments with a closed-form model for the calculation of individual fracture modes. Among other anticrack phenomena, the collapse of weak snow layers is a present and tangible example. To enhance our understanding of the fracture behavior of porous materials under mixed-mode loading involving both closing mode I and mode II, we introduce a modification of the conventional fracture mechanical experiment PST. This methodology provides insight into previously unexplored fracture regimes. The design has the advantage that the anticrack is confined to the weak layer. As a result, the mode mixity of the crack-tip loading remains fairly constant during crack growth because the anticrack cannot kink. A similar geometry was used for some of the first measurements of supersonic shear cracks[33], a phenomenon recently observed for anticracks confined in weak snow layers[19,34,35]. Because of the confinement of the anticrack, we measure fracture toughness in terms of critical energy release rates $\mathcal{G}_c$ rather than stress intensity factors $K_i$, which take on complex values for interfacial cracks.

Since fracture processes are driven by the global energy balance of structures, structural models are needed to interpret the experimental results. To analyze PST data, we consider a closed-form analytical solution for a first-order, shear-deformable, layered plate under cylindrical bending (slab) that is supported by an elastic foundation (weak layer)[36]. The anisotropy induced by the layered nature of snow slabs is accounted for by a stiffness matrix that distinguishes extension, bending, shear, and bending–extension coupled deformations. Here, we extend the model to consider additional surface loads as used in the present experimental setup (cf. "Methods"). Mode I and II energy release rates are calculated from the stresses and displacements in the weak interface. The model was validated with comprehensive numerical studies[18,36], full-field displacement measurements[31], and dynamic energy release during anticrack growth[37]. Crucial inputs are the elastic material properties of snow slab and weak layer. For this purpose, we parameterized the elastic modulus of slab layers using density measurements and tested for the sensitivity of the model with respect to parameter assumptions (cf. "Methods").

Our tailored mixed-mode fracture test (MMFT) is based on the classical PST. Specific modifications allow us to measure the critical energy release rate over the entire range of mode interactions and, thus, allow for in situ fracture toughness measurements. The method consists of extracting a 100 cm long and 30 cm wide rectangular block from the snowpack containing an 11.5 cm thick slab layer, a weak layer, and a 6 cm thick base layer. This snow block is then mounted on a tilting device to perform MMFTs at different inclinations (Fig. 1a, b). Extracting a slab of defined thickness allows for controlling its stiffness. By adding variable surface dead loads in the form of steel bars, we can then control the cut length a priori to increase the mode II contribution. Artificial anticracks are introduced by pushing the back of a snow saw into the weak layer, and upon reaching the critical cut length, the anticrack propagates causing the entire weak layer to collapse. This instability point defines the critical energy release rate derived from

the model described above. Cutting can be performed in upslope or downslope direction of the tilted MMFT. Note that in most PST experiments thus far, cutting was in the upslope direction (Fig. 1c, e). Additional details on the experimental procedures are given in the Supplementary Material.

The proposed modifications have the following advantages. A thinner slab (1) reduces the effect of slab layering, (2) decreases the bending stiffness that can cause strong stress concentrations in the weak layer, (3) more closely resembles the slender beam geometry of our theory, (4) minimizes impeding bending at the slab normal faces, and (5) allows for transporting and tilting the fragile samples as most of the slab load is removed. Tilting the snow block up to 65° allowed us to cover the entire range of mixed-mode loading scenarios. Overall, our MMFT experiments enabled us to perform 88 experiments on two weak layers (cf. "Methods"). The reproducibility of the results permits generalized statements of the mixed-mode fracture properties of weak snow layers.

Here, we show that fracture toughness values are significantly larger in shear than in collapse, and suggest a power-law interaction between the anticrack propagation modes.

## Results

### Cut lengths increase with slope angle

Introducing an artificial crack into the weak layer modifies the global energy balance of the system. Changes in the total potential energy per unit crack advance define the energy release rate (ERR). At the critical cut length $a_c$ (Fig. 1a), the critical energy release rate is reached, marking the onset of unstable anticrack growth. Hence, $a_c$ is closely related to fracture toughness. Yet, it must not be misinterpreted as a material property, as it is influenced by many system variables including cutting direction, slope angle, slab layering, and load[38]. Changes in such system variables are reflected in the scatter of recorded $a_c$ values (Fig. 1c). The data show that increasing the surface loads at site B from 1.48 kN/m² to 2.53 kN/m² (extra load) reduces measured critical cut lengths considerably although the same slab and weak layer were tested. Additional details are given in the "Methods" section.

In our data (Fig. 1c, green), it is evident that $a_c$ increases with slope angle $\varphi$. Previous experiments were inconclusive, suggesting either a decrease[39], an increase[40,41], or no discernible trend[42] for $a_c$ as function of slope angle. The contradictory results can be attributed to PST recording standards[43] that recommend upslope cuts. In addition, most weak layers were tested on slope angles below 36°[41]. In this regime (−45° to 0° in Fig. 1c), our results (green) are also inconclusive and mostly influenced by changes in system variables (e.g., loading and layering). The compilation of historic PST data[44] (Fig. 1c, orange) shows no trend in slope angle because many different weak layers and slab assemblies were tested with upslope cuts below −36°. The influence of the slope angle on $a_c$ is much more pronounced when cutting in downslope direction (>0° in Fig. 1c), confirming recent theoretical considerations of the differences in shear loading of upslope and downslope cuts (see "Methods" section)[36].

### Weak-layer anticracks are controlled by fracture toughness

The energy balance of a weak-layer anticrack at the critical cut length, i.e., the energy release rate (ERR) at which it becomes unstable, is a fundamental material property and known as its fracture toughness. In the present setup, the total energy release rate $\mathcal{G} = \mathcal{G}_I + \mathcal{G}_{II}$ comprises closing mode (I) and shearing mode (II) contributions. Using the virtual crack closure technique allows for the identification of the ERR of a given crack length and the separation of both contributions[36].

For the $a_c$ values recorded in our experiments, Fig. 1d shows computed critical shearing mode ERRs ($\mathcal{G}_{II}$) as a fraction of the corresponding total ERRs ($\mathcal{G} = \mathcal{G}_I + \mathcal{G}_{II}$). Because the weak-layer fracture toughness is a material property, all measurements collapse onto one curve despite varying structural conditions. In particular, the data recorded at site B, with extra load (2.53 kN/m²) and considerably

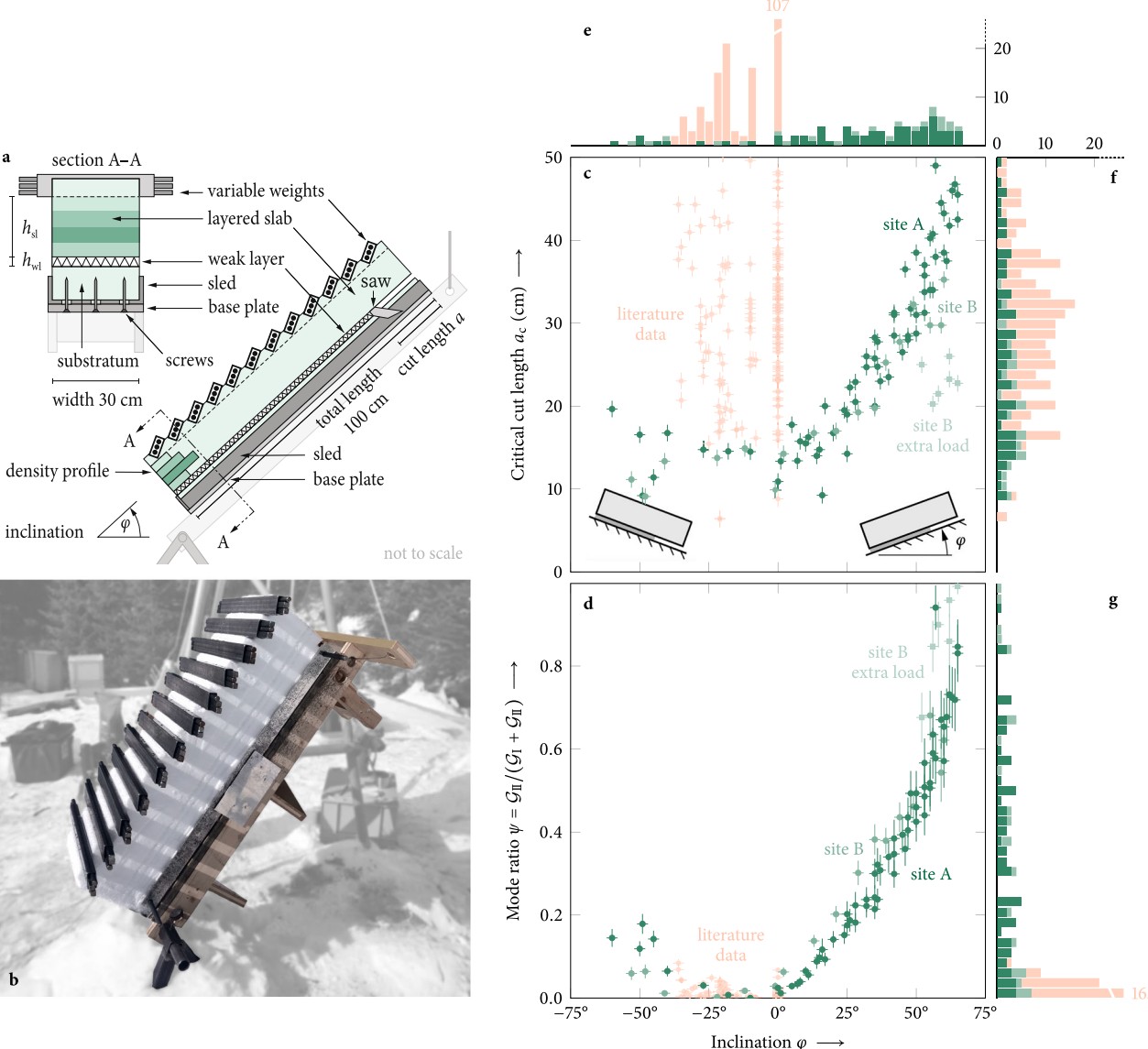

**Fig. 1 | Mixed-mode fracture tests. a** Illustration of the experimental setup. **b** Extracted slab–weak-layer assembly with added weights at 60° prior to cutting the weak layer. **c** Critical cut lengths values, the measured length at which the artificially introduced crack becomes unstable, for the present study (green, $|S| = 88$, where $S$ is the set of experimental samples) and from the literature[44] (orange, $|S| = 183$). The literature dataset contains only upslope cuts ($\varphi \leq 0°$) but several different weak layers and slab assemblies such that recorded cut lengths scatter widely. Tests on the same slab–weak-layer assembly with constant added weights (site A), 1.59 kN/m², show a trend of increasing critical cut lengths $a_c$ with inclination for downslope cuts ($\varphi > 0°$). Increasing surface dead loads (site B) with extra load, 2.53 kN/m², vs. site B, 1.48 kN/m², breaks the observed trend. Critical cut lengths were measured with ±1 cm uncertainty and slope angles with ±2° uncertainty. **d** Mode II energy release rate at the onset of unstable crack propagation as a fraction of the total energy release rate. The mode II fraction increases rapidly with inclination for downslope cuts ($\varphi > 0°$) but only moderately for upslope cuts

($\varphi < 0°$). All data points (site A, site B, site B with extra load, literature data) follow the same trend with little scatter. The literature dataset comprises almost no mode II contribution, even at inclinations as high as 36°. Mode-ratio uncertainties were calculated from error propagation of uncertainties in cut length (±1 cm), inclination (±2°), and weak-layer thickness (±1 mm). **e–g** Stacked histograms truncated at 20 counts per bin. The distribution of tested inclinations (**e**) shows that, historically, propagation saw tests were predominantly performed in flat terrain and cut upslope (orange) while the present study (green) focuses on steep downslope cuts. Critical cut lengths (**f**) are distributed uniformly with mean and standard deviation of $a_c = 31.0 \pm 14.4$ cm indicating equal likelihood for a wide range of cut lengths. The distribution of mode II fractions of the total energy release rate (**g**) shows that the literature data (orange) contains no information on mode II energy release rates while the present study (green) covers the full range between pure mode I and pure mode II fracture toughness of weak layers.

shorter cut lengths (Fig. 1c, light green), collapse onto the same curve (Fig. 1d). Changes in the slab's structural parameters (e.g., extra loading), that generally introduce considerable scatter in recorded critical cut lengths (Fig. 1c), have much less impact on the ratio of computed critical ERRs. This highlights that fracture toughness is the crucial descriptor of weak-layer anticrack propagation and an essential input for predictions about fracture processes that precede slab avalanche release.

Our data show a pronounced asymmetry between upslope ($\varphi < 0°$) and downslope ($\varphi > 0°$) cuts (Fig. 1d)[45]. While downslope cuts permitted measurements of pure mode II crack propagation (at $\varphi \approx 65°$), PSTs with upslope cuts at the same inclination ($\varphi = -65°$) are still dominated by mode I (more than 70% of the total ERR). For this reason, none of the historical PST experiments measured mode II contributions (Fig. 1g). Note that pure mode II crack propagation required steep inclinations (up to $\varphi = 65°$), downslope cuts, and in

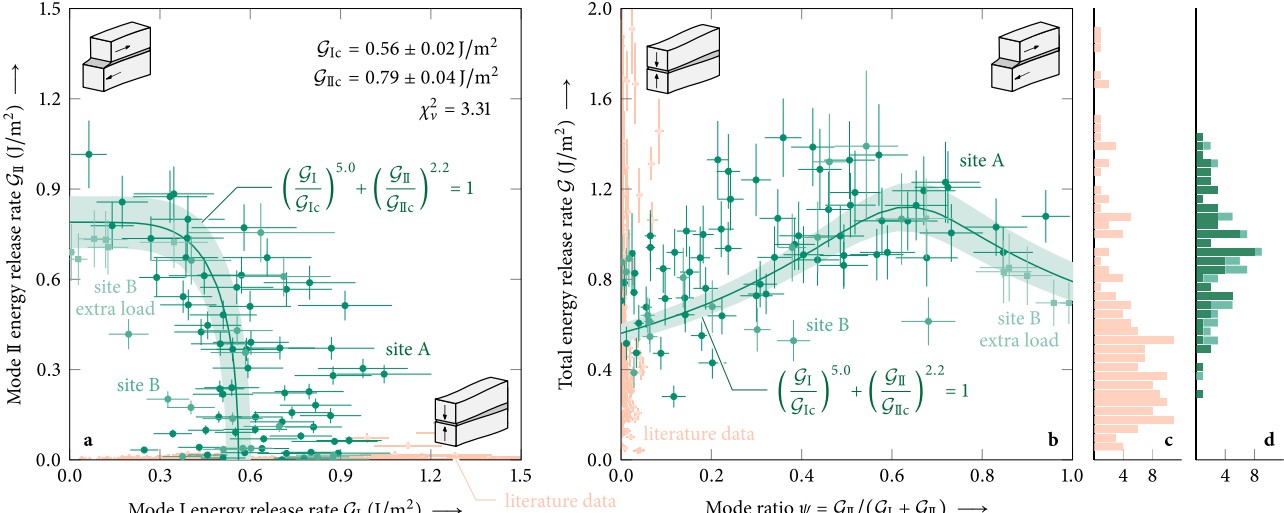

**Fig. 2 | Mixed-mode interaction law of weak-layer fracture toughness. a** Mode I/ II composition of critical energy release rates at the onset of unstable crack propagation in the surface-hoar weak layer at field site A (Fig. 3) with 1.59 kN/m² added surface load (|S| = 65 samples, Feb 18 to Mar 3, 2022), field site B (Fig. 3) with 1.48 kN/m² (|S| = 17, Mar 7 to 9, 2022), and field side B with 2.53 kN/m² (|S| = 6, Mar 10, 2022). The mixed-mode interaction law is determined from an orthogonal distance regression ($p < 0.001$) and shown with 95% confidence bands. **b** Total energy release rate $\mathcal{G} = \mathcal{G}_I + \mathcal{G}_{II}$ as a function of mode ratio $\psi$ (mode II fraction), observed in our data (|S| = 88, green) and the literature dataset[44] (|S| = 183, orange).

Because of diverse snowpack conditions, the literature dataset (orange) scatters widely. Notably, it cannot provide information on mode II failure. In the present dataset (green), we observe a maximum of the total energy release rate at $\psi \approx 0.6$, although data for $\psi > 0.6$ is scarce. It is evident that it is not the total energy release rate that governs crack propagation, but the interaction law given in Eq. (1). **c** Histogram of recorded total energy release rates in literature[44] with mean total critical energy release rate $\bar{\mathcal{G}}_c = 0.98 \pm 0.02$ J/m² and median $\tilde{\mathcal{G}}_c = 0.53 \pm 0.06$ J/m². **d** Histogram of recorded total energy release rates in the present study with mean $\bar{\mathcal{G}}_c = 0.90 \pm 0.01$ J/m² and median $\tilde{\mathcal{G}}_c = 0.90 \pm 0.10$ J/m².

particular significant surface loads (2.53 kN/m²). Counterintuitively, pure mode I anticrack propagation is not observed in flat-field tests ($\varphi = 0°$) but for upslope cuts between $\varphi = -5°$ and $-15°$. Both the slope-angle asymmetry and the offset of pure mode I anticrack propagation have their origin in the combined compression and shear loading of the weak layer.

The weak layer is subjected to different sources of normal and shear deformations arising from the gravitational forces exerted by the slab and supplementary weights. These forces precipitate (i) the settlement of the slab, (ii) the generation of moments due to the eccentricities in their lines of action, and (iii) the creation of moments during the cutting process. Both categories of moments induce bending at the slab ends, yet they differ significantly in their characteristics and magnitudes. The superposition of these effects, incorporating both normal and tangential components, is responsible for the asymmetric behavior of mode II contributions to the total energy release rate (ERR), particularly with respect to variations in slope inclination and cutting direction (Fig. 1d). Additional details are given in the "Methods" section.

The predominance of mode I in upslope cuts is due to the synergistic effects of eccentricity-induced bending (ii) and cut-induced bending (iii), which increase compression in the weak layer at the lower end of the slab, even at steep inclinations where normal settlement (i) becomes negligible. Conversely, eccentricity-induced bending (ii) causes an uplift at the upper end of the slab, stretching the weak layer as the inclination increases, while the effect of cut-induced bending (iii) reduces. Consequently, mode I vanishes at high inclinations, and a mode II dominated regime forms below $\varphi = 90°$. At low angles, a pure mode I state is not observed at $\varphi = 0°$, since the shear component due

to bending dominates at small inclinations. This component enhances crack-tip shear deformations in downslope cuts and reduces these deformations in upslope cuts[36,46].

We observe that our MMFTs, together with the model, allow us to derive energy release rates of weak-layer anticracks in pure mode I, pure mode II, and all mode interactions in between.

## Mixed-mode interaction law for weak snow layers
In most real-world cases, the total energy release rate is composed of contributions from different modes, i.e., mixed-mode loading, and rarely from one pure fracture mode. In these cases, the resistance of a material against crack growth is captured by so-called mixed-mode interaction laws, describing the limit of stable crack propagation for all load states between the pure fracture modes. Using an orthogonal distance regression[47], we determined a mixed-mode interaction law for weak-layer anticracks, in the form of a power law[22]

$$\left(\frac{\mathcal{G}_I}{\mathcal{G}_{Ic}}\right)^{\frac{1}{n}} + \left(\frac{\mathcal{G}_{II}}{\mathcal{G}_{IIc}}\right)^{\frac{1}{m}} = 1, \tag{1}$$

from our field data on two weak layers consisting of buried surface hoar (Fig. 2a, Table 1). The power-law exponents $n, m \in [0, 1]$ are metrics for the interaction of both fracture modes where $n, m \longrightarrow 0$ corresponds to independent failure modes and $n, m = 1$ to a very strong interaction (see Supplementary Material for explanations of data fitting procedures). The total fracture toughness $\mathcal{G}_c = \mathcal{G}_{Ic} + \mathcal{G}_{IIc}$ is comparable to other field data on layers of surface hoar[44,48] (0.33 ± 0.17 J/m² and 0.1 to 1.5 J/m², respectively), and ice–aluminum[49] (1 J/m²) or ice–steel[49] (5 J/m²) interfaces. However, note that in the present case, fracture is not controlled by total fracture toughness $\mathcal{G}_c = \mathcal{G}_{Ic} + \mathcal{G}_{IIc}$, but by the interaction law given in Eq. (1). This is evident in the distinct maximum of the total energy release rate at $\psi \approx 0.6$ when plotted against the mode ratio $\psi$ (Fig. 2b).

The observed ratio of $\mathcal{G}_{IIc}$ to $\mathcal{G}_{Ic}$ is consistent with many other materials whose mode II fracture toughnesses are larger than their mode I counterparts[22]. This result is remarkable because

## Table 1 | Fracture toughness of weak snow layers

| Weak layer type | $\mathcal{G}_{Ic}$ (J/m²) | $\mathcal{G}_{IIc}$ (J/m²) | $n$ | $m$ |
|---|---|---|---|---|
| Surface hoar | 0.56 ± 0.02 | 0.79 ± 0.04 | 0.20 | 0.45 |

Weak-layer fracture toughness of surface hoar for collapse (mode I) and in-plane shear (mode II) fracture with interaction-law shape parameters, cf. Eq. (1).

macroscopically, mode I in our experiments corresponds to collapse rather than tensile failure. In solid materials, mode I cracks can only grow under tensile loads. Certainly, in highly porous materials, microscopic failure is dominated by tensile mixed-mode I/II fracture of the solid matrix, e.g., owing to the bending of surface-hoar ice crystals[50]. Here, we expect the dominance of mode II over mode I, i.e., $\mathcal{G}_{\text{IIc}} > \mathcal{G}_{\text{Ic}}$[22]. Our data suggest that the dominance of mode II translates to the macroscopic scale of highly porous materials under compressive loads. Equally noteworthy is the fact that our data show that superimposing compression and shear weakens the material in terms of fracture (crack propagation). This is again consistent with most other materials but remarkable because superimposing compression and shear were shown to reinforce weak layers with respect to their apparent strength (initial failure)[51]. We attribute this to the porous and low-density microstructure of weak layers. On intact weak layers, compressive stresses can cause bond breaking at the microscale that is invisible at the macroscale if the applied stresses are smaller than the macroscopic compressive strength[32]. The micro defects compact and initially strengthen the weak layer with respect to shear loading. As a consequence, higher superimposed compressive loads cause a more abrupt and violent subsequent shear failure[32]. In the presence of a crack, the effect is not beneficial because both compression and shear loading increase the stored energy and, hence, the energy release rate. That is, both facilitate crack growth. It is important to note that the fracture toughness measured here is a macroscopic property. It encompasses all micromechanical effects and does not distinguish individual microscopic effects such as bond breaking or friction.

## Discussion

### Implications and limitations for avalanche modeling

Using a nonlocal mechanical model for the global energy balance at the onset of anticrack growth, we estimated fracture toughness values for weak snow layers under a wide range of mixed-mode loading conditions, from pure shear (mode II) to pure compression (mode I). Results from in situ experiments showed that modeled fracture toughness values followed a power-law interaction, with estimated critical energy release rates in mode I lower than in mode II. While our results provide the first mixed-mode interaction law for the fracture toughness of highly porous weak snow layers, our conclusions are limited as experiments were performed solely on buried surface hoar. For a complete picture of snow weak-layer fracture, data for other types of weak layers, such as faceted crystals or depth hoar, are needed. Furthermore, since the size and shape of the interaction law are influenced by the choice of elastic-modulus parametrization, future experiments should include video recordings to better estimate the elastic properties from measured displacement fields[31]. Although we performed 88 experiments, the number of measurements at high inclinations is still limited. This is mostly because these measurements are more challenging, resulting in larger measurement uncertainties (error bars) and broader confidence intervals in the pure mode II regime.

Other porous materials are often described using similar power-law type interaction laws, with either equal or unequal exponents to capture the relationships between stress intensity factors under mixed loading conditions[52,53]. Common predictors for fracture toughness across highly porous materials include density[54] and microstructure[55,56], where higher density generally correlates with increased fracture toughness[57]. Despite the extensive literature on the tensile and bending properties of porous materials, studies focusing on compressive fracture properties remain scarce, highlighting a pervasive challenge in understanding fracture behaviors under compressive loads[53,56]. The concept of anticracks under compressive mode I loading has been explored in man-made materials like glass foams[3] and 3D-printed brittle open-cell structures[5], and studies indicate that the morphology of anticracks under compression resembles tensile mode I cracks[3]. Nevertheless, there is a notable absence of experiments measuring mixed compression and shear fracture

toughness across various materials, especially natural porous materials such as snow.

The shape of the interaction law is different from the stress-based failure envelope where shear strength is generally lower than compressive strength[51], yet $\mathcal{G}_{\text{Ic}}$ is smaller than $\mathcal{G}_{\text{IIc}}$. The common assumption that avalanches release more easily on steeper terrain[58] is primarily based on avalanche observations and our understanding of snow strength. Certainly, the higher likelihood for avalanches on steeper terrain can be justified by factors such as snow friction angle. However, our results indicate that not all factors that govern the avalanche release process increase avalanche likelihood with slope angle. The relevant observation would be whether it is easier to trigger a so-called whumpf, the large-scale collapse of a weak layer, in a flat field or an avalanche in a steep slope, albeit the entire snow cover is the same. Yet, no such field study is available. Indeed, our results show that $\mathcal{G}_{\text{Ic}}$ is smaller than $\mathcal{G}_{\text{IIc}}$, suggesting that anticrack propagation may encounter less resistance in low-angle terrain. This may partly explain the enormous distances sometimes observed in the remote triggering of slab avalanches[59].

Crack propagation is accompanied by singular crack-tip stress concentrations and governed by energy alone[60]. Crack nucleation, however, originating from weakly singular and nonsingular stress concentrations, is governed by both stress and energy simultaneously. This is reflected in virtually all modern fracture mechanics methods, such as finite fracture mechanics[61], or phase-field models for fracture[62]. Because both strength and toughness are involved, the weak layer's low shear strengths can cause early initial failure on steep slopes. This concerns not only the slope inclination but also to the angle at which skiers load the snowpack, e.g., combined compression and shear in turns. It is to be noted that weak layer strengths have only been examined in one study and significantly more data is needed[51]. Ideally, strength and fracture tests of the same weak layer should be performed simultaneously for a conclusive picture.

Holistic computational models that use, e.g., the material-point method (MPM) or the discrete-element method (DEM), are increasingly used to investigate the dynamics of anticrack propagation in snow and avalanche release[19,63,64]. While such models can shed light on internal processes that cannot readily be measured, they are almost exclusively validated against the stress-based failure envelope[32,63]. The present data provide an additional benchmark that can be used to verify that both strength and toughness are represented correctly and in accordance with the physics of weak layer failure.

The present work represents the first measurement of a compression–shear fracture-toughness interaction law derived from modeling and experiments of anticracks in highly porous materials and for weak snow layers of surface hoar. Its relevance extends beyond snow and concerns, e.g., porous rocks that may form compaction bands[12], porous seams in sedimentary rocks that develop pressure dissolutions[65], or brittle foams used as sandwich core materials in composite materials that dominantly transfer shear loads[3]. For snow avalanches, the interaction law observed in our data is very relevant as it provides fracture toughness values that could serve as crucial ingredients for predictive analytical tools[18] and to assess avalanche hazards based on modeled snow cover parameters.

## Methods

### Field site and snowpack

All experiments were performed between February 18 and March 10, 2022, on a flat and uniform site near Davos, Switzerland (Fig. 3), located on the roof of two buildings in a forest opening protected from wind. Most experiments were performed on the roof of building A, and after it was cleared of snow, we also carried out experiments on building B. The presence of a nearby creek, the absence of direct sunlight in winter, and the cold concrete roof (typically below 0 °C) create favorable conditions for the formation and preservation of

surface hoar. Both weak layers tested consisted of surface hoar, buried by a snowfall at the beginning of January 2022, with a mean weak layer thickness of 9.02 mm. We characterized the snowpack with a manual snow profile[66] (Fig. 4), and measured snow density with a 50 cm³ cylindrical density cutter (23 mm inner diameter). Additional details on the field site and snowpack are given in the Supplementary Material.

## Mixed-mode fracture tests

To measure fracture toughness over the entire range of mode interactions, from pure mode I to pure mode II, we performed mixed-mode

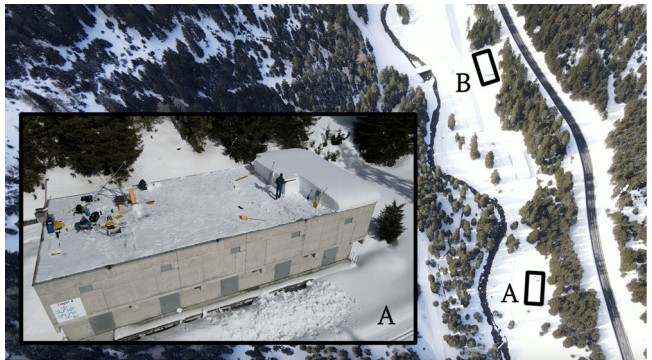

**Fig. 3 | Field site.** Bird's eye view of the geographic location of field sites A and B with a closer look at the roof of the building at site A.

fracture tests (MMFTs), modified propagation saw tests (PST). Our experiments consisted of extracting snow blocks (1000 mm long and 300 mm wide) containing a weak layer from the snowpack. This was done using an aluminum sled that we pushed into the snowpack below the weak layer of interest and isolating the snow block with snow saws. We then reduced the slab above the weak layer to a thickness of 150 mm before making serrated cuts, slanted at an angle of 65°, on the surface of the slab using a sharp device. The mean slab thickness from the weak layer to the bottom of the serrated cuts was then 115 mm. The snow block and sled were then mounted on a tilting device, consisting of a wooden base plate with guiding rails on each sidewall and two rows of 40 mm long screws that penetrated the substratum below the weak layer through holes in the sled. The guiding rails and the screws ensured that the snow block would not slide off when tilted at steep angles. On one end, the base plate was fixed to the ground on a pivot point. On the opposite end, the base plate was attached with a steel cable to a tower made of scaffolding poles, allowing us to tilt the base plate to any desired angle.

MMFTs were then performed at different angles after 12 weights were placed on the slab (Fig. 1a, b). Each weight consisted of a rectangular hollow steel profile with up to three metal rods. Each component (profile and rod) weighed up to 1 kg, allowing for different load levels. An artificial cut was then introduced in the weak layer by pushing the unserrated back of a snow saw (2 mm thickness) into the weak layer. At the time the crack propagated and the weak layer collapsed, the critical cut length from saw tip to slab face was measured on both sidewalls and averaged when the cutting was not perfectly

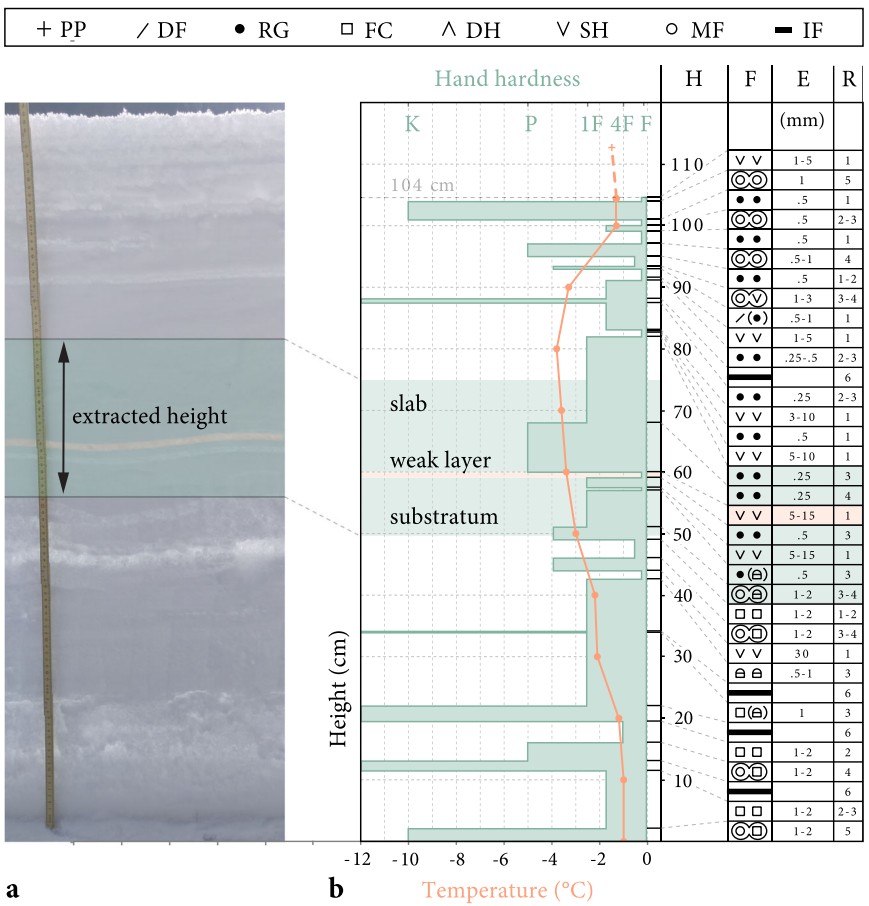

**Fig. 4 | Snow profile. a** Isolated column of the snowpack at field site A on March 4, 2022, with the extracted section highlighted in green and the weak layer in orange. **b** Corresponding manual snow profile with hand hardness index (bar length), snow temperature (orange line), distance from ground (H), grain type (F, see legend), grain size (E), hand hardness (R). Grain types indicated in the figure legend are precipitation particles (PP), decomposing and fragmented precipitation particles (DF), rounded grains (RG), faceted crystals (FC), depth hoar (DH), surface hoar (SH), melt forms (MF), and ice formations (IF)[66].

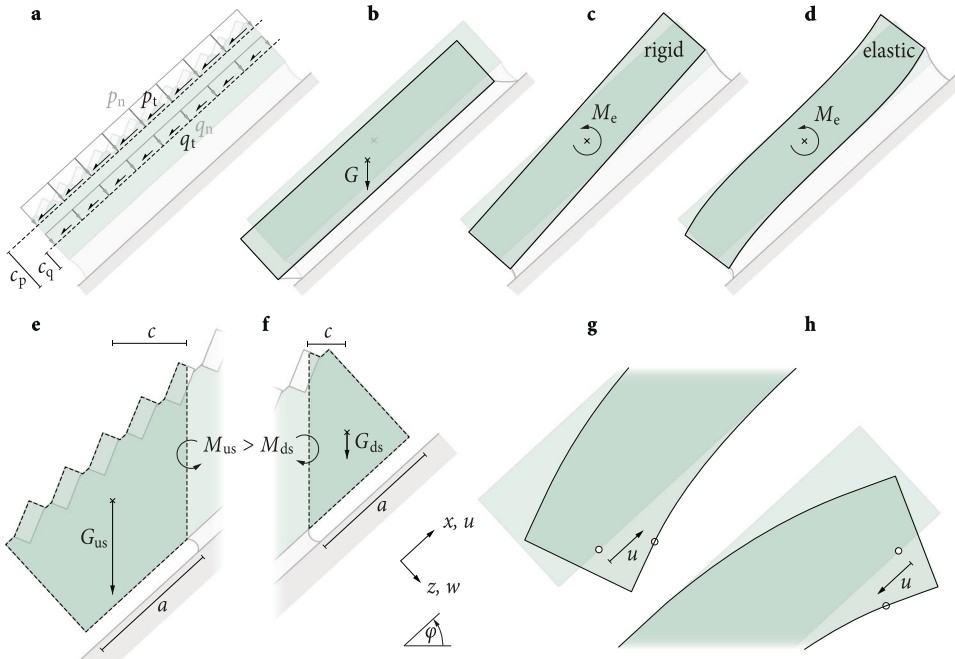

**Fig. 5 | Specimen asymmetries due to cutting direction and inclination. a** Acting forces, represented by line loads of the slab density $q$ and the added dead weights $p$, and shown with slope-normal ($x$, n) and slope-parallel tangential ($z$, t) components. The effective lines of action (dashed) have slope-normal distances of $c_q$ and $c_p$, respectively, from the weak layer. **b** The gravitational pull $G$ on the total mass ($q + p$) causes a slope-normal and slope-parallel settlement of the slab. **c** For a rigid slab, the moment $M_e$ induced by the eccentricities $c_q$ and $c_p$ compresses the weak layer at the lower end and expands it at the upper end. **d** With increasing slab compliance, slab deformations concentrate toward the slab ends. **e** Upslope (us) cuts induce additional bending moments $M_{us}$ due to the load $G_{us}$ of unsupported slab segments. **f** Downslope (ds) cuts generate smaller bending moments ($M_{ds}$) as the load ($G_{ds}$) is smaller. **g** Slab bending induces upward shear deformations ($u$) at the lower slab end. **h** At the upper slab end, these deformations point downward.

perpendicular. Additional details on the experimental procedures are given in the Supplementary Material.

**Asymmetry in energy release rates**

In our experimental configuration, there is an asymmetry in the mode II contributions to the overall energy release rate (ERR) with slope inclination and cutting direction. This asymmetry arises from the sample geometry, the loading configuration, and the material compliance, which affect the translation of the slab as well as the rotational and bending moments in the slab (Fig. 5).

In our experiments, a rectangular snow beam under its own weight $q$ and additional surface loads $p$ is inclined at an angle $\varphi$ (Fig. 5a). The gravitational pull on the snow beam ($G$ in Fig. 5b) induces both slope-parallel shearing and slope-normal settlement in the weak layer. Slope-parallel displacement increases from zero on flat terrain ($\varphi = 0°$) to a maximum as $\varphi$ approaches 90°, while slope-normal displacement decreases from its peak at $\varphi = 0°$ to zero at $\varphi = 90°$.

Even without a cut in the weak layer, there is a rotational moment $M_e$ in the slab, compressing the weak layer at the lower end of the beam and stretching it at the upper end (Fig. 5c). This is due to the eccentricities of the lines of action of the slab weight $q$ and the added surface weights $p$ relative to the weak layer ($c_q$ and $c_p$ in Fig. 5a). This effect intensifies with increasing $\varphi$ and is concentrated toward the slab ends for compliant slabs (Fig. 5d).

When a cut is introduced in the weak layer, the system configuration changes as a part of the beam becomes unsupported (Fig. 5e, f). An additional bending moment is then introduced in the slab, which is larger for an upslope cut ($M_{us}$, Fig. 5e) than for a downslope cut ($M_{ds}$, Fig. 5f). This is because the unsupported slab segment, and thus the corresponding gravitational load ($G$), is larger for upslope cuts than for downslope cuts. Slab bending introduces both slope-parallel shearing and slope-normal settlement in the weak layer. While there is weak

layer compression for both cut directions, it is larger for upslope cuts due to the greater load. The sign of the slope-parallel shearing, on the other hand, depends on the cutting direction. For upslope cuts, slab bending counteracts gravitational shearing (Fig. 5g), while for downslope cuts, it enhances gravitational shear deformations (Fig. 5h).

The superposition of these three effects—translation, rotation, and bending—leads to the asymmetry in mode II contributions to the total ERR observed in our experiments (Fig. 5d). For "upslope cuts," all three effects compress the weak layer, and slab bending counteracts gravitational shearing. As a result, the ERR for upslope cuts is predominantly driven by mode I, with only a marginal increase in the mode II ratio as slope inclination rises, thus preventing a pure mode II condition even at $\varphi = −90°$. In fact, slab bending is the reason why pure mode I anticrack propagation is not observed in flat-field tests ($\varphi = 0°$) but occurs for upslope cuts between $\varphi = −5°$ and $−15°$. For "downslope cuts," translation and slab bending compress the weak layer while eccentricity-induced bending moments lift the slab off the weak layer, an effect that increases with $\varphi$. At the same time, slab bending enhances gravitational shearing. A localized absence of compressive normal contributions therefore occurs when compression and lifting effects offset each other, leading to a state of pure mode II ERR below $\varphi = 90°$ for downslope cuts.

**Mechanical model**

We extend the closed-form analytical model for the mechanical response stratified snowpacks of Weißgraeber and Rosendahl[36] by adding surface loads. For this purpose, we modify the equilibrium conditions, Eqs. (6a)–(6c),

$$0 = \frac{dN(x)}{dx} + \tau(x) + q_t + p_t, \tag{2a}$$

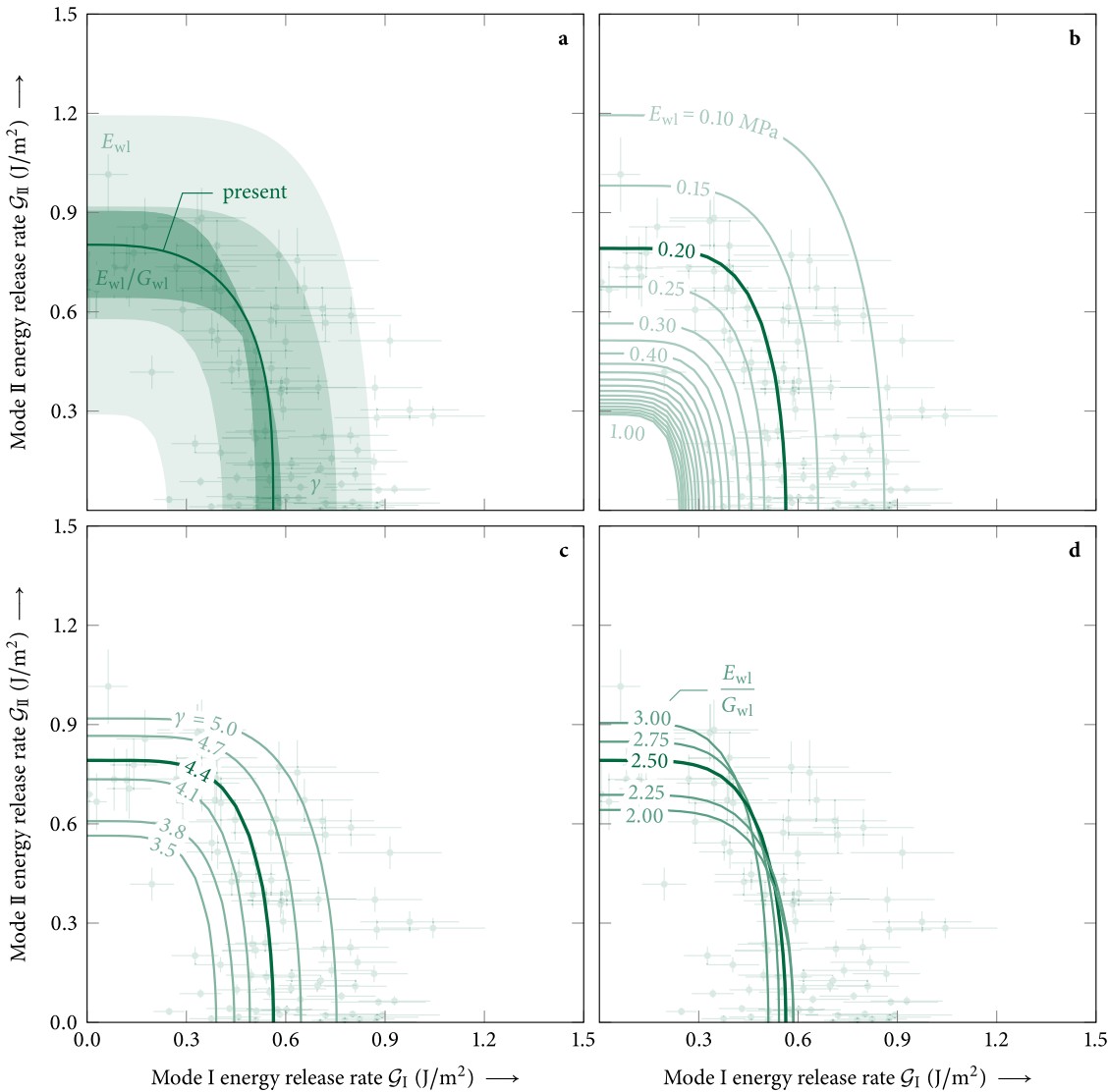

**Fig. 6 | Mixed-mode fracture toughness sensitivity to elastic properties.**
**a** Changes in the fracture toughness with different elastic properties. **b** Sensitivity to the elastic modulus of the weak layer $E_{wl} \in [0.1, 1.0]$ MPa[48]. **c** Sensitivity to the exponent $\gamma \in [3.5, 5.0]$[48] of the density parametrization of the elastic modulus of slab layers (6). **d** Sensitivity to the ratio of elastic and shear modulus of the weak layer $E_{wl}/G_{wl} \in [2.0, 3.0]$ representing Poisson ratios $\nu \in [0.0, 0.5]$ between no lateral expansion and incompressibility, respectively.

$$0 = \frac{dV(x)}{dx} + \sigma(x) + q_n + p_n, \tag{2b}$$

$$0 = \frac{dM(x)}{dx} - V(x) + \frac{h+t}{2}\tau(x) + z_s q_t - \frac{h}{2}p_t, \tag{2c}$$

by adding normal and tangential surface loads $p_n$ and $p_t$, where $x$ is the axial coordinate, $N$ and $V$ are normal and transverse section forces, $M$ is the bending moment, $\sigma$ and $\tau$ are the weak layer's compressive and shear stresses, $q_n$ and $q_t$ are the normal and tangential components of the slab's weight load, $h$ and $t$ are the thicknesses of slab and weak layer, and $z_s$ the $z$-coordinate of the center of the slab's gravity. This changes Eq. (A10), the vector

$$\mathbf{d} = \left[0, q_t + p_t, 0, q_n + p_n, 0, z_s q_t - \frac{h}{2}p_t\right]^{\top}, \tag{3}$$

of supported slab segments and as a consequence Eq. (13), the particular integral

$$\mathbf{z}_p = \left[\frac{q_t + p_t}{k_t} + \frac{h(h+t-2z_s)q_t}{4\kappa A_{55}} \quad 0 \quad \frac{q_n + p_n}{k_n} \quad 0 \quad \frac{(2z_s - h - t)q_t + (2h+t)p_t}{2\kappa A_{55}} \quad 0\right]^{\top}, \tag{4}$$

but leaves stiffness matrices and the general solution unchanged. Similarly, Eq. (B14), the vector

$$\mathbf{p}(x) = \begin{bmatrix} -\frac{q_t+p_t}{2A_{11}}x^2 - \frac{B_{11}}{6K_0}(q_n+p_n)x^3 \\ -\frac{q_t+p_t}{A_{11}}x - \frac{B_{11}}{2K_0}(q_n+p_n)x^2 \\ -\frac{A_{11}}{24K_0}(q_n+p_n)x^4 \\ -\frac{A_{11}}{6K_0}(q_n+p_n)x^3 \\ \frac{A_{11}}{6K_0}(q_n+p_n)x^3 + \left(z_s - \frac{B_{11}}{A_{11}}\right)\frac{q_t}{\kappa A_{55}} - \frac{hp_t}{2\kappa A_{55}} - \frac{q_n+p_n}{\kappa A_{55}}x \\ \frac{A_{11}}{2K_0}(q_n+p_n)x^2 - \frac{q_n+p_n}{\kappa A_{55}} \end{bmatrix}, \tag{5}$$

of unsupported segments is adjusted where $A_{11}$, $B_{11}$, $D_{11}$, and $\kappa A_{55}$ are the slabs laminate stiffness and $K_0 = B_{11}^2 - A_{11}D_{11}$. This allows for the consideration of added weights, in the present case in the form of steel rods, at the slab's surface while leaving solution procedure described by Weißgraeber and Rosendahl[36] unchanged. Added weights can be represented by distributed surface loads because, owing to St. Venant's principle, their effect on weak-layer stresses and deformations is equivalent to concentrated loads in sufficient distance from the load application point. Additional details on the derivation of the governing equations are given in the Supplementary Material.

## Elastic modulus parametrization and parameter sensitivity

The elastic properties of the slab and especially the weak layer are crucial input parameters for the mechanical model. As we do not have direct measurements, we used parametrizations and literature values and examined the sensitivity of the model with regard to assumptions made.

The elastic modulus of slab layers, consisting of rounded grains, was calculated from

$$E_{sl}(\rho) = E_0 \left( \frac{\rho}{\rho_0} \right)^\gamma, \tag{6}$$

where $\rho_0 = 917\ \text{kg/m}^3$ is the density of ice and $E_0 = 6.5 \times 10^3\ \text{MPa}$ is the elastic modulus of ice[67–69]. The exponent $\gamma = 4.4$ was determined from the elastic response observed in flat-field experiments[48]. For the weak layer, consisting of buried surface hoar, we assumed a density of $\rho_{wl} = 100\ \text{kg/m}^3$, Young's modulus of $E_{wl} = 0.2\ \text{MPa}$, and Poisson's ratio of $\nu = 0.25$[70], corresponding to a ratio between elastic modulus and shear modulus of $E_{wl}/G_{wl} = 2.5$.

To evaluate the sensitivity of the model with regard to the above assumptions, we investigated the impacts of the exponent $\gamma$ of the density parametrization, the weak-layer elastic modulus $E_{wl}$, and the ratio between elastic and shear modulus $E_{wl}/G_{wl}$ of the weak-layer (Fig. 6). Physically meaningful parameter ranges are $\gamma \in [3.5, 5.0]$, $E_{wl} \in [0.1, 1.0]$ MPa, and $E_{wl}/G_{wl} \in [2.0, 3.0]$. This yields elastic moduli of slab layers between 1 and 400 MPa for densities between 150 and 400 kg/m³. While all parameters have some influence on the magnitude of the energy release rates, results remain in the same order of magnitude and the principal shape of the interaction law is mostly unaffected (Fig. 6). The weak-layer elastic modulus has the largest influence (Fig. 6a, b). Changes in elastic properties ($E_{wl}$, $\gamma$) affect the magnitude of the total energy release rate without affecting the mode I/II ratio $\mathcal{G}_{Ic}/\mathcal{G}_{IIc}$ (Fig. 6b, c). Even the ratio of weak-layer elastic modulus to shear modulus has only a small impact on the mode I to mode II ratio $\mathcal{G}_{Ic}/\mathcal{G}_{IIc}$ (Fig. 6d). Overall, uncertainties from the elastic parameters used as input for the model are on the same order of magnitude as measurement uncertainties from the cut length $\Delta a = \pm 10$ mm, weak-layer thickness $\Delta t = \pm 1$ mm, and slope angle $\Delta \varphi = \pm 2°$.

## Data availability

The dataset including data processing routines is available under Creative Commons Attribution 4.0 International license from https://doi.org/10.5281/zenodo.11443644[71]. Source data are provided with this paper.

## Code availability

A Python implementation of the mechanical model is publicly available from the code repository https://github.com/2phi/weac or for direct installation from https://pypi.org/project/weac (last accessed January 31, 2024)[72]. Routines used for data processing are part of the dataset (see "Data availability").

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

## Acknowledgements
We thank Henning Löwe for his computer-tomography analyses of our weak-layer samples. We are grateful for the support of Moritz Altenbach in collecting the field data. This work was in part supported by grants from the Swiss National Science Foundation (200021_169424 and 200021L_201071) and funded by the Deutsche Forschungsgemeinschaft (DFG, German Research Foundation) under grant no. 460195514.

## Author contributions
P.W., A.H., and P.L.R. conceived the study. V.A., B.B., and A.H. developed the experimental method. P.W. and P.L.R. developed the theoretical framework. V.A., B.B., and A.H. collected the field data. All authors contributed to the interpretation of the results and writing of the manuscript.

## Funding

## Competing interests
The authors declare no competing interests.
