## [Peer Review File · Nature Communications]

Fracture toughness of mixed-mode anticracks in highly porous materialsREVIEWER COMMENTS

Reviewer #1 (Remarks to the Author):

See attached document

Reviewer #1 (Remarks on code availability):

I cloned a local version of the code and was able to run it just fine. It appeared to have replicated the results exactly. The code is well documented and easy to follow. My only comment is that there is no README file with instructions for installing and running the code. I'm familiar with Docker and was able to reproduce the environment, but some reviewers/users may not be.

As a side note, Ocean Code only allows users with academic emails to make an account and run the code on their site. This requires reviewers like myself to download the code and sort out the dependencies/environment ourselves. I think this is where a fleshed-out README would help.

Reviewer #2 (Remarks to the Author):

In this paper a modification of the propagation saw test (PST) for weak layers in snow is proposed, a large experimental campaign is developed, and its results are carefully analysed by modelling mixed-mode anticrack propagation in a weak layer (collapsed weak layer) of porous materials. Such modification of PST is applied to determine the mixed-mode fracture toughness of surface hoar buried by a snowfall. The PST data are studied considering an analytic formulation based on first-order shear deformation theory of laminate plates under cylindrical bending for the snow slab with the Winkler elastic foundation model (spring model) for the weak layer of surface hoar. The article represents a relevant and novel contribution to the problem of snow avalanche triggering by providing a new mixed-mode (compression+shear) interaction law for weak snow layers. Thus, it is recommended for publication after taking into account the following minor comments:

Article:

1) The authors should clarify in the abstract and in the main article text that they are determining the mixed mode fracture toughness for a weak layer of highly porous materials subjected to shear and compression, not the mixed mode fracture toughness of a material itself. This makes a great difference, because an anticrack (or crack) propagating in a weak layer is trapped therein and can propagate in a mixed mode. However, a crack in a bulk material subjected to a mixed model usually kinks changing its direction of propagation.

2) Figure 1. Some additional explanations could be useful. For example: What is the parameter N ? What do the sites A and B mean? How is the (anti)crack length obtained? Do you know where the (anti)crack is located a priori? How did you obtain these results?

3) In Fracture Mechanics the concepts of fracture toughness (K_{Ic}) and fracture energy= critical energy release rate (G_{Ic}) of a material are under certain conditions equivalent but not the same. Nevertheless, sometimes G_{Ic} is referred to fracture toughness. This issue deserves at least a comment.

4) Is there any physically based motivation for proposing an interaction power law in terms of the components of the critical ERR with different exponents? I am asking because in a quite comprehensive review of closely related interaction laws for fracture energy in spring (Winkler) models by Mantic et al (Int. J. Fract., 2015, Vol 195, pages 15–38), and apparently also in the literature cited therein, only power laws with equal exponents are considered. Have you tried to perform fitting of your experimental results by considering the same exponents?

3) Figure 4. If some symbols related to the snow profile are included, the mechanical engineering community might not understand the meaning. Some brief comments or reference to the list of such symbols would be useful.

4) Lines 415-423. Please refer to Figure 1 where the specimen and test configuration is represented. More references would be helpful.

5) Mechanical model. If N is the normal force in beam model, then the parameter N in Figure 1 should be denoted by another letter or this notation issue clarified.

Supplementary material

In this document the authors provide additional information about the experiments and the analytical and numerical model applied. The first part is dedicated to experiments, explaining how, where and when specimens were obtained. The calculation of the snow density and the critical ERR is also described. The second part justifies the model applied. This document is very well written, so only a few minor comments are given:

1) General comment. Sometimes it is difficult to follow the explanations in the article and supplementary material and more cross-references to equations and sections would be very helpful. For example, in section "Data fitting procedure" the vector β is not explained, it is described later in section "Interaction-law identification" and also in the main manuscript. Some cross-references could make it clearer.

2) General comment. It seems that it is not mentioned in the article that there is supplementary material. A comment would be helpful (in the same way that codes availability is mentioned).

3) Section "Identification-law" verification: The objective of this section is not clear. Are you comparing r_1 and r_2 ? Are you applying both?

4) Section "Finite element model": a few details are given about the numerical model. How do you define the weak layer elements? Are you using cohesive zone model for example?

5) Section "Model Validation": Although it is mentioned, the explanations in this section are closely related to those in References [11-13]. The authors should clarify the novelty of contributions of this part of the present work regarding the previous ones, including other apparently simpler and less general proposal by Chiaia et al. (Cold Regions Science and Technology, 2008, Vol 53, pages 170-178).

6) Regarding the cited successful applications of finite fracture mechanics approaches with weak-interface models, the seminal contribution by Cornetti et al (Int J Solids Struct, 2012, 49, pages 1022-1032) also could be mentioned.

Reviewer #3 (Remarks to the Author):

The title and abstract of the submitted paper promise to talk about highly porous materials and, in particular, their fracture toughness. The abstract asserts to present new experimental tests to found (to measure? to calculate?) fracture mechanics proprieties in natural snowpack. Then, using a mechanical model to interpret the data (what data measured in the field? Densities?), the authors calculate the fracture toughness for anticrack growth from pure shear to pure collapse (I think I did not understand very well the use of the word "collapse" in this paper ...). However, the paper is entirely dedicated to the presentation of new and (debatable) experimental and numerical results of a single type of snow (surface hoar). There are no comparisons/discussions of the results with other types of highly porous materials. This is both from an experimental and theoretical point of view.

In fact, porosity is not a protagonist of the paper. Nor are porous materials. For this reason alone, the reader is disappointed with the content of the paper compared to the expectations provided by the title.

The authors write an introduction (overview) briefly summarising the bibliography on porous media, fracture mechanics, and then give a lot of space to fracture mechanics and related experimental tests on snow material.

It is not clear why this work is also mentioned:

Kiakojouri, F., De Biagi, V., Chiaia, B. & Sheidaii, M. R. Progressive collapse of framed building structures: Current knowledge and future prospects. *Engineering Structures* 206, 110061 (2020). URL <https://www.sciencedirect.com/science/article/pii/S0141029619322576>.

It is true that the authors state that the mechanism of snowpack collapse is progressive, but the cited paper:

- 1) is a review of the state-of-art on progressive collapse on frame structures (reinforced concrete, steel,...);
- 2) it is a comparison of the various robustness techniques of framed building structures;
- 3) it discusses pure and mixed progressive collapse mechanisms, therefore very far from the subject of the submitted paper.

The overview mentions the experimental PST tests (used to measure of fracture properties in snow, exclusively) and it proposes MMFT test for, I think, the measurement of fracture toughness of the (snow) weak-layer in mixed mode.

The MMFT experimental test is presented here for the first time. It is my opinion that authors must be very careful with respect to the parameter that they want and think to measure (which may not coincide).

Surely, the paper is the result of serious and remarkable work:

- 1) The effort to design and implement a new experimental test to understand the of slab avalanche release;
- 2) the development of a numerical model;
- 3) the extensive experimental campaign.

In my opinion, the work will not be significant for the field and related fields. The theory used (the model) is already presented and known (also) in the snow and, unfortunately, the data (and not the parameters calculated with the numerical model) from the extensive experimental campaign (the real novelty of the article) may not be used to validate other models as they are flawed by the imposed overload.

The paper does not present any noteworthy novelties compared to the established literature, (among which the following are missing:

Ritter, J., Löwe, H. & Gaume, J. Microstructural controls of anticrack nucleation in highly porous brittle solids. *Sci Rep* 10, 12383 (2020). <https://doi.org/10.1038/s41598-020-67926-2>

Heierli, J., Gumbsch, P. & Zaiser, M. Anticrack nucleation as triggering mechanism for snow slab avalanches. *Science* 321, 240–243. <https://doi.org/10.1126/science.1153948> (2008).

Mulak, D. & Gaume, J. Numerical investigation of the mixed-mode failure of snow. *Comput. Part. Mech.* 6, 439–447. <https://doi.org/10.1007/s40571-019-00224-5> (2019)), ,

but only a different (numerical) analysis of the mixed mode fracture toughness of the weak-layer proposed in the form of power-law (Eq. 1), mimicking classical fracture mechanics.

The work supports its claims, but these are as trivial as "Cut lengths increase with slope angle".

With regard to experimental tests and subsequent analysis/interpretation, I do not understand the need to carry out an experimental campaign with a new test (with even additional loads) when unconventional snow fracture tests (PST) have recently been validated by the scientific community. Furthermore, I do not understand the reason for the imposed overload and its way to apply (why a distributed load and not a point load?). For the spontaneous release of snow avalanches, the 'load' of the slab above the weak layer would have been sufficient ...

Instead of the mixed-mode interaction law for weak-layer anticracks, it would have been more interesting to have the influence of gravity (and inclination) with respect to the in-plane GI and GII or directly a formula (like the classical fracture mechanics formulas) that gives the G (or K) dependent on the specimen size, crack length and/or other measurable quantities.

Although 'energetic', the methodology is based on the theory of laminated plates under cylindrical bending with the weak-layer modelled as an elastic (Winkler) foundation, a typically tensional model (into which stiffnesses enter). From this, using a simple Mohr-Coulomb resistance criterion, the authors calculate the normal and tangential stresses (at the crack tip) and, consequently, the

respective $G = G_I + G_{II}$.

Therefore, G or G_I or G_{II} are not calculated directly as for standard tests (e.g., 3PBT), but calculated by means of a numerical tension model.

For those of us in classical fracture mechanics, it is a big difference between G 'energy release rate' and K (or K_c) 'fracture toughness', which are linked together by the modulus of elasticity. I would ask the authors to do a thorough check to see if the fracture toughness they indicate is not instead an energy release rate. I would ask the authors to report the quantities with their symbol from classical fracture mechanics.

The methods are detailed enough to allow the work to be reproduced.

The method is "circular": the authors tries to validate a numerical model that attempts to reproduce an experiment in the field with the aim is to determine an unmeasurable physical quantity. In my opinion, the process is too tweakable, risking that the physical quantities involved in the model lose their own physical meaning.

Reviewer #4 (Remarks to the Author):

Review: Fracture toughness of mixed-mode anticracks in highly porous materials

Submitted to *Nature Communications*

March 2024

1 General

In this paper, the authors develop and describe a new experimental approach for studying the mixed-mode failure mechanisms of porous materials. The methodology is applied to weak layers in stratified snowpacks, and their experimental approach allows them to measure a much wider range of loading states than was previously reported in the literature. The authors formulate an interaction law between pure compression and pure shear in the form of a power law, which was fit to their experimental data. They illustrate in the supplementary sections that this power law formulation fits the data better than a previous method. These experimental results are supplemented with mechanical modeling to investigate the sensitivity of the phenomenon to different snow elastic properties. As noted by the authors, these findings have implications for modeling avalanche onset, as well as other porous materials. As a snow modeler, I appreciate that this work provides another useful benchmark for validating models that try to capture snow's failure behavior. I look forward to seeing results from more snow types in the future.

The science seems solid, and the new experimental approach seems like an advancement of the state of the art for in-situ snow toughness measurements. Overall, the document is well written, interesting, and presented in an accessible way. I have asked questions below to clarify some details and to better understand the data fitting procedure, which some of the main conclusions (i.e. $\mathcal{G}_{IIIc} > \mathcal{G}_{Ic}$) hinge upon.

2 Specific Comments - Manuscript

1. Line 125 - Is “mixed-mode” supposed to be repeated twice?
2. Line 144-145 - The phrasing of “...ii) decreases the bending stiffness causing stronger stress concentrations in the weak layer...” tripped me up a few times. I believe the authors are trying to say that the bending stiffness causes stronger stress concentrations in the weak layer, and that reducing the slab thickness reduces the effect of that bending stiffness. But this

sentence almost reads as if the *reduction* in the bending stiffness is making *larger* stress concentrations in the weak layer. It's a subtle difference in verbiage, but I suggest rephrasing as "...ii) decreases the bending stiffness **that can cause strong** stress concentrations in the weak layer..."

3. Lines 215-238 - It could be beneficial to add a diagram (if space allows) to supplement the description of the shear loading in the weak layer. I had a hard time visualizing the force and moment balances the authors were describing, and attempted to draw a diagram myself in order to try and understand it. I feel this could help other readers, too. A diagram may also help me understand how an inclination of 65 degrees corresponds to pure shear fracture when it is still under transverse loads from gravity (lines 208-209)?
4. Line 210 - What is the 70% relating to? Is it the percentage of upslope PST results that were all mode I? Or something else?
5. Fig. 1d - Are the error bars in the modeled ϕ output related to those in the measurements in 1c.? Or does the model provide uncertainty values as an output? I ask because the error bars on the measurements in 1c seem fairly constant across inclination angle, whereas the ϕ error bars seem to increase with inclination.
6. Fig 2a - Can you comment on how "good" a goodness-of-fit value of 3.31 is? I want to make sure I'm correctly interpreting the result of Eqn S6. What is the range of these values (good to bad)? There appears to be more data points along the x-axis of Fig. 2a that are closer to \mathcal{G}_I 0.8-0.9+ range than there are for \mathcal{G}_{II} along the y-axis. I realize the orthogonal distance regression factors in the data uncertainty, but it's difficult to discern the uncertainty bars on some of the points, and therefore whether the \mathcal{G}_{Ic} intercept makes sense.
7. Line 344 - The last " ," should be a "."
8. Between lines 349-350 - "need" is used twice. Consider removing the first instance.
9. Line 800 - "Because of **divers** snowpack conditions..." I believe this should be "Because of **diverse** snowpack conditions..."
10. The findings in "Elastic modulus parametrization and parameter sensitivity" were not revisited in the Discussion to highlight their significance. The results of this study were presented, and then kind of just dropped. Can anything be gleaned from those findings for future experiments, or follow on model development?

3 Specific Comments - Supplementary Information

1. Lines 056-062 - Is there any specific guidance on what minimum stiffness is needed to guide future experiments?
2. Between lines 149-150 - Can you comment on whether cut speed affects the critical length? Is 70 mm/s a fast/normal/slow cut speed for PSTs?
3. Line 209 - “The cut length is recorded **form** the end of the saw...” I believe this should be “**from**” instead.
4. Lines 1337-1339 - Can you expand on what you mean by the last sentence in this paragraph? Specifically, what is meant by “...without further resolving their microscopic nature.” Although you may be referring to crack propagation models specifically, there are examples of recent micro-mechanical models in the snow literature (discrete element method, peridynamics) that can replicate microstructures from micro-CT scans, and then simulate the microscopic behavior of those snow grains and the fracture of sintered bonds between them.

Revision of NCOMMS-24-10218

Fracture toughness of mixed-mode anticracks in highly porous materials

Dear Editors and Reviewers,

Thank you for your thorough review and constructive feedback on our manuscript. We appreciate the opportunity to revise our work based on your comments and suggestions. We carefully considered each point and made corresponding revisions to enhance the clarity, accuracy, and impact of our manuscript.

In this document, we address each remark (*blue, italic*) point-by-point (black) and detail the changes we have made to the manuscript (*italic quotes*). We have attached a revised manuscript file with changes highlighted. We hope that these modifications adequately address the concerns raised and improve the manuscript. We are grateful for the guidance that the reviewers' expertise has provided and are confident that these changes have strengthened our submission.

Reviewer #1 (main text)

Thank you for taking the time to carefully read our manuscript and point out the following typing errors:

Remark 1.1

Line 125 - Is "mixed-mode" supposed to be repeated twice?

Line 344 - The last ",," should be a "."

Between lines 349-350 - "need" is used twice. Consider removing the first instance.

Line 800 - "Because of divers snowpack conditions..." I believe this should be "'Because of diverse snowpack conditions...'"

Line 209 - "The cut length is recorded form the end of the saw..." I believe this should be "from" instead.

We appreciate the attention to detail. We thoroughly reviewed the manuscript and corrected all identified typos to enhance the clarity and accuracy of the text.

Remark 1.2 *Line 144-145 - The phrasing of "...ii) decreases the bending stiffness causing stronger stress concentrations in the weak layer..." tripped me up a few times. I believe the authors are trying to say that the bending stiffness causes stronger stress concentrations in the weak layer, and that reducing the slab thickness reduces the effect of that bending stiffness. But this sentence almost reads as if the reduction in the bending stiffness is making larger stress concentrations in the weak layer. It's a subtle difference in verbiage, but I suggest rephrasing as "...ii) decreases the bending stiffness that can cause strong stress concentrations in the weak layer..."*

We have rephrased the paragraph according to your suggestion as follows:

A thinner slab i) reduces the effect of slab layering, ii) decreases the bending stiffness that can cause strong stress concentrations in the weak layer, iii) more closely resembles the slender beam geometry of our theory, iv) minimizes impeding bending at the slab normal faces, and v) allows for transporting and tilting the fragile samples as most of the slab load is removed.

Remark 1.3 *Lines 215-238 - It could be beneficial to add a diagram (if space allows) to supplement the description of the shear loading in the weak layer. I had a hard time visualizing the force and moment balances the authors were describing, and attempted to draw a diagram myself in order to try and understand it. I feel this could help other readers, too. A diagram may also help me understand how an inclination of 65 degrees corresponds to pure shear fracture when it is still under transverse loads from gravity (lines 208-209)?*

Thank you for this suggestion. We have added Fig. 5 including the following clarification in the Methods section:

Asymmetry in energy release rates

In our experimental configuration, there is an asymmetry in the mode II contributions to the overall energy release rate (ERR) with slope inclination and cutting direction. This asymmetry arises from the sample geometry, the loading configuration, and the material compliance, which affect the translation of the slab as well as the rotational and bending moments in the slab (Fig. 5).

In our experiments, a rectangular snow beam under its own weight q and additional surface loads p is inclined at an angle φ (Fig. 5a). The gravitational pull on the snow beam (G in Fig. 5b) induces both slope-parallel shearing and slope-normal settlement in the weak layer. Slope-parallel displacement increases from zero on flat terrain ($\varphi = 0^\circ$) to a maximum as φ approaches 90° , while slope-normal displacement decreases from its peak at $\varphi = 0^\circ$ to zero at $\varphi = 90^\circ$.

Even without a cut in the weak layer, there is a rotational moment M_e in the slab, compressing the weak layer at the lower end of the beam and stretching it at the upper end (Fig. 5c). This is due to the eccentricities of the lines of action of the slab weight q and the added surface weights p relative to the weak layer (c_q and c_p in Fig. 5a). This effect intensifies with increasing φ and is concentrated towards the slab ends for compliant slabs (Fig. 5d).

When a cut is introduced in the weak layer, the system configuration changes as a part of the beam becomes unsupported (Fig. 5e,f). An additional bending moment is then introduced in the slab, which is larger for an upslope cut (M_{us} , Fig. 5e) than for a downslope cut (M_{ds} , Fig. 5f). This is because the unsupported slab segment, and thus the corresponding gravitational load (G), is larger for upslope cuts than for downslope cuts. Slab bending introduces both slope-parallel shearing and slope-normal settlement in the weak layer. While there is weak layer compression for both cut directions, it is larger for upslope cuts due to the greater load. The sign of the slope-parallel shearing, on the other hand, depends on the cutting direction. For upslope cuts, slab bending counteracts gravitational shearing (Fig. 5g), while for downslope cuts, it enhances gravitational shear deformations (Fig. 5h).

*The superposition of these three effects — translation, rotation, and bending — leads to the asymmetry in mode II contributions to the total ERR observed in our experiments (Fig. 5d). For **upslope cuts**, all three effects compress the weak layer, and slab bending counteracts gravitational shearing. As a result, the ERR for upslope cuts is predominantly driven by mode I, with only a marginal increase in the mode II ratio as slope inclination rises, thus preventing a pure mode II condition even at $\varphi = -90^\circ$. In fact, slab bending is the reason why pure mode I anticrack propagation is not observed in flat-field tests ($\varphi = 0^\circ$) but occurs for upslope cuts between $\varphi = -5^\circ$ and -15° . For **downslope cuts**, translation and slab bending compress the weak layer while eccentricity-induced bending moments lift the slab off the weak layer, an effect that increases with φ . At the same time, slab bending enhances gravitational shearing. A localized absence of compressive normal contributions therefore occurs when compression and lifting effects offset each other, leading to a state of pure mode II ERR below $\varphi = 90^\circ$ for downslope cuts.*

Fig. 5 | Specimen asymmetries due to cutting direction and inclination. **a** Acting forces, represented by line loads of the slab density q and the added dead weights p , and shown with slope-normal (x, n) and slope-parallel tangential (z, t) components. The effective lines of action (dashed) have slope-normal distances of c_q and c_p , respectively, from the weak layer. **b** The gravitational pull G on the total mass ($q + p$) causes a slope-normal and slope-parallel settlement of the slab. **c** For a rigid slab, the moment M_e induced by the eccentricities c_q and c_p compresses the weak layer at the lower end and expands it at the upper end. **d** With increasing slab compliance, slab deformations concentrate towards the slab ends. **e** Upslope (us) cuts induce additional bending moments M_{us} due to the load G_{us} of unsupported slab segments. **f** Downslope (ds) cuts generate smaller bending moments (M_{ds}) as the load (G_{ds}) is smaller. **g** Slab bending induces upward shear deformations (u) at the lower slab end. **h** At the upper slab end, these deformations point downwards.

Remark 1.4 Line 210 - What is the 70% relating to? Is it the percentage of upslope PST results that were all mode I? Or something else?

This refers to the mode I fraction of the total energy release rate. We have added the following clarification:

While downslope cuts permitted measurements of pure mode II crack propagation (at $\varphi \approx 65^\circ$), PSTs with upslope cuts at the same inclination ($\varphi = -65^\circ$) are still dominated by mode I (more than 70% of the total ERR).

Remark 1.5 Fig. 1d - Are the error bars in the modeled φ output related to those in the measurements in 1c? Or does the model provide uncertainty values as an output? I ask because the error bars on the measurements in 1c seem fairly constant across inclination angle, whereas the φ error bars seem to increase with inclination.

The error bars given in Figs. 1c and 1d are in both cases the measurement uncertainty, which we assume to be $\pm 2^\circ$. This uncertainty then serves as an input for the error propagation into mode ratios and energy release rates shown in Figs. 1d and 2. We have added a clarification on this in the captions of Figs. 1c and 1d:

Critical cut lengths were measured with ± 1 cm uncertainty and slope angles with $\pm 2^\circ$ uncertainty. Mode-ratio uncertainties were calculated from error propagation of uncertainties in cut length (± 1 cm), inclination ($\pm 2^\circ$), and weak-layer thickness (± 1 mm).

Remark 1.6 *Fig 2a - Can you comment on how “good” a goodness-of-fit value of 3.31 is? I want to make sure I’m correctly interpreting the result of Eqn S6. What is the range of these values (good to bad)?*

This is a good question, as identifying an appropriate goodness-of-fit measure is very challenging for scattered data. A reduced chi-squared value χ_v^2 close to 1 generally indicates that the model fits the data well, where <1 indicates overfitting and >1 underfitting. It accounts for both the number of degrees of freedom and error variances. In theory, a reduced chi-squared of 3.31 thus indicates a relatively poor fit between model and data. However, for datasets with intrinsic scatter beyond measurement errors—such as snow science and fracture mechanics where individual variation is significant—the model is not expected to predict each point accurately. In these cases, a higher reduced chi-squared value can be acceptable. If the residuals (difference between observed and model-predicted values) are randomly distributed (no pattern), the fit still represents the data reasonably well, despite a chi-squared value >1 . Systematic patterns would suggest model inadequacies, which are not present. χ_v^2 then becomes particularly relevant in the comparison of different models against each other (e.g. Fig. 2 vs. Fig. S5). Here, a lower $\chi_v^2 > 1$ is considered better. We have added a comment on this to the “Data fitting procedure” section in the supplement:

A χ_v^2 value close to 1 generally indicates that the model fits the data well, where < 1 indicates overfitting and > 1 underfitting. For datasets with intrinsic scatter beyond measurement errors—such as snow science and fracture mechanics where individual variation is significant—the model is not expected to predict each point accurately. In these cases, a higher χ_v^2 value can be acceptable if residuals are randomly distributed and do not exhibit patterns.

Remark 1.7 *There appears to be more data points along the x-axis of Fig. 2a that are closer to GI 0.8-0.9+ range than there are for GII along the y-axis. I realize the orthogonal distance regression factors in the data uncertainty, but it’s difficult to discern the uncertainty bars on some of the points, and therefore whether the GIc intercept makes sense.*

That fact that the dataset contains more data points close to the x-axis than close to the y-axis in Fig. 2a is reflected in the narrower 95%-confidence bands closer to the x-axis. Obtaining more data points close to the y-axis is the fundamental challenge in our experiments, and unfortunately we were not able to collect more data points in this region. The Code Capsule provided alongside this manuscript and Tables S4 and S5 contain the raw data, including uncertainties. This allows for an in-depth analysis of the G_{Ic} intercept.

Remark 1.8 *The findings in “Elastic modulus parametrization and parameter sensitivity” were not revisited in the Discussion to highlight their significance. The results of this study were presented, and then kind of just dropped. Can anything be gleaned from those findings for future experiments, or follow on model development?*

Thanks for this suggestion. These results are included to provide a more comprehensive understanding of our study. In snow and avalanche science such parametrizations are commonly used and often debated. The sensitivity study shows that our interpretations and findings do not substantially change when the parametrization changes. We acknowledge that we did not discuss any practical implications. We therefore added the following to the discussion:

Furthermore, since the size and shape of the interaction law are in part influenced by the choice of elastic-modulus parametrization, future experiments should be recorded on video to directly estimate the elastic properties from measured displacement fields.¹

Reviewer #1 (supplementary materials)

Remark 1.9 *Lines 056-062 - Is there any specific guidance on what minimum stiffness is needed to guide future experiments?*

Securing the substratum to prevent sliding remains a significant challenge in these experiments, particularly on steep inclines. About 70% of our experiments failed for inclinations greater than

60°. The primary difficulty was finding a good trade-off between the number of screws required to prevent substrate sliding against the need to not disturb the structure of the snowpack too much. We found that thick screws are more effective than nails at bearing transverse loads. For future experiments, we plan to further optimize this, for instance by using aluminum fins rather than screws.

Regarding substrate properties, we determined that a minimum density of approximately 250 kg/m³ was necessary, and the snow should consist of small grain types (e.g. rounded grains or decomposing and fragmented precipitation particles). In contrast, snow consisting of larger faceted crystals or melt-freeze crusts posed greater challenges, as the substratum then disintegrated more readily when pushing it down on the screws.

Your feedback has prompted us to clarify that our reference to 'stiffness' might be misleading. It is not mechanical stiffness per se, but rather a sufficient level of strength within the substratum that is crucial. We have revised our text accordingly to better reflect this nuance:

"The substratum needed a particular level of cohesion or bonding strength, usually found in small grains with a density exceeding 250 kg/m³, to support the snow block during tilting. Similarly, the slab had to be easily profiled while also supporting additional weights."

Remark 1.10 *Between lines 149-150 - Can you comment on whether cut speed affects the critical length? Is 70 mm/s a fast/normal/slow cut speed for PSTs?*

The cutting process introduces a time-dependent component to the critical cut length and the energy release rate observed in our experiments. Time-dependent deformations within the slab may lead to the development of delayed stresses. It is crucial to acknowledge that the analytical model used in this study does not account for time-dependent dissipative processes, and thus may not fully capture these effects. Nevertheless, existing research indicates that the time dependency of Propagation Saw Test (PST) results remains negligible across variations in cutting speed, whether slow (< 8 mm/s) or fast (> 260 mm/s)². Our cutting speed was typical for PSTs.

Furthermore, we assume that any changes in the slab properties over time due to additional loading are sufficiently reflected by the density measurements recorded immediately after the experiment. It is unlikely that significant changes in the properties of the weak layer occur due to added load. Studies have shown that the specific fracture energy of weak layers remains consistent under additional load for the time scales involved in our experiments (< 1 hour), emphasizing that fracture toughness is an inherent characteristic of the material³.

We have added the following note to the Experimental procedure section in the supplement:

The cutting speed employed here is typical for conducting propagation saw tests. Note that variations in cutting speed between 8 mm/s and 260 mm/s do not significantly affect the critical cut length.²

Remark 1.11 *Line 209 - "The cut length is recorded from the end of the saw..." I believe this should be "from" instead.*

Good catch, thank you! We have corrected the typo in the manuscript.

Remark 1.12 *Lines 1337-1339 - Can you expand on what you mean by the last sentence in this paragraph? Specifically, what is meant by "...without further resolving their microscopic nature." Although you may be referring to crack propagation models specifically, there are examples of recent micro-mechanical models in the snow literature (discrete element method, peridynamics) that can replicate microstructures from micro-CT scans, and then simulate the microscopic behavior of those snow grains and the fracture of sintered bonds between them.*

This is a good point. While our model uses the fracture mechanics perspective of macroscopic (smeared) fracture properties, recent studies provided insights into the effects of the microstructure. To reflect this, we have extended the discussion and added references to recent studies in that direction:

Effective quantities of fracture mechanics models always include microscopic mechanisms without further resolving their microscopic nature.⁴ Identifying microstructural interactions in the fracture of highly-porous materials is a current active research topic.⁵⁻⁷

Reviewer #2 (main text)

Remark 2.1 1) *The authors should clarify in the abstract and in the main article text that they are determining the mixed mode fracture toughness for a weak layer of highly porous materials subjected to shear and compression, not the mixed mode fracture toughness of a material itself. This makes a great difference, because an anticrack (or crack) propagating in a weak layer is trapped therein and can propagate in a mixed mode. However, a crack in a bulk material subjected to a mixed model usually kinks changing its direction of propagation.*

Thank you for the pertinent remark. This point is indeed important, and we clarified it in the manuscript. We agree that the mechanics of interface cracks and those confined to thin layers are different from those in the bulk. For instance, the stresses at crack tips in both bulk materials and interfaces decay with $1/\sqrt{r}$ with r being the radial distance from the crack tip. While the singularity exponent is $\lambda = 1/2$ in the bulk, it becomes complex in the interface $\lambda = 1/2 + \gamma i$, where γ is the so-called bi-material constant and a function of the elastic properties of the adjacent materials. Yet, in both cases, mode mixity is governed by the local crack-tip loading conditions, captured by the stress intensity factors K_I , K_{II} (and K_{III}). Because typically, K_{Ic} is smaller than K_{IIc} or K_{IIIc} (least resistance against crack growth), bulk material cracks have a preference to kink out of mixed-mode conditions into pure mode I crack-tip loading.

Cracks confined in interfaces or thin layers, cannot escape the mixed-mode loading conditions. This has experimental advantages, as not only initial crack propagation, but the entire crack extension phase occurs under the same (or at least very similar) mixed-mode conditions, which makes observations of the mixed-mode fracture toughness much easier. A famous study has exploited this fact.⁸ The authors bonded two pieces of Homalite-100, a brittle polyester resin using temperature-enhanced surface sintering and milled a notch along the bondline. This way, they were able to confine cracks within the interface while investigating a single-material specimen. Using projectile-induced mode-II loading, they were able to show that mode II cracks can travel faster than the shear wave speed. Since mode-I cracks cannot travel at these speeds,⁹ it is evident that they actually observed mode-II cracks that did not kink. Similar observations have recently been made for cracks confined in weak snow layers.^{10,11} Similarly, our theory supports that weak-layer cracks are subject to mixed-mode loading even when—or rather specifically because—they cannot escape the combined shear and compression loading.

On another note, the specific microstructure of weak-layers is only present in confinement. One could argue that measuring a weak-layer bulk fracture toughness is pointless as they are only present as thin layers in nature.

We have added the following discussion to the manuscript.

To enhance our understanding of the fracture behavior of porous materials under mixed-mode loading involving both closing mode I and mode II, we introduce a modification of the conventional fracture mechanical experiment PST. This methodology provides insight into previously unexplored fracture regimes. The design has the advantage that the anticrack is confined to the weak layer. As a result, the mode mixity of the crack tip loading remains fairly constant during crack growth because the anticrack cannot kink. A similar geometry was used for some of the first measurements of supersonic shear cracks,⁸ a phenomenon recently observed for anticracks confined in weak snow layers.¹⁰⁻¹² Because of the confinement of the anticrack, we measure fracture toughness in terms of critical energy release rates \mathcal{G}_c rather than stress intensity factors K_i , which take on complex values for interfacial cracks.

Remark 2.2 2) *Figure 1. Some additional explanations could be useful. For example: What is the parameter N ? What do the sites A and B mean? How is the (anti)crack length obtained? Do you know where the (anti)crack is located a priori? How did you obtain these results?*

Thank you for this suggestion. We have added the following clarifications to the caption of Fig. 1: i) that $|S|$ (previously N) is the cardinality of the set S of experimental samples, i.e., refers to the number of samples, ii) that cuts, i.e., anticracks, are introduced into the weak layer, iii) that the critical cut length is obtained as the length at which the artificially introduced anticrack becomes unstable, and iv) have added a cross reference explaining sites A and B. The caption now reads:

Fig. 1 | Mixed-mode fracture tests. **a** Illustration of the experimental setup. **b** Extracted slab–weak-layer assembly with added weights at 60° prior to cutting the weak layer. **c** Critical cut lengths values, the measured length at which the artificially introduced crack becomes unstable, for the present study (green, $|S| = 88$, where S is the set of experimental samples) and from the literature² (orange, $|S| = 183$). The literature data set contains only upslope cuts ($\varphi \leq 0^\circ$) but several different weak layers and slab assemblies such that recorded cut lengths scatter widely. Tests on the same slab–weak-layer assembly with constant added weights (site A (Fig. 4), 1.59 kN/m) show a trend of increasing critical cut lengths a_c with inclination for downslope cuts ($\varphi > 0^\circ$). Increasing surface dead loads (site B (Fig. 4) with extra load, 2.53 kN/m , vs. site B, 1.48 kN/m) breaks the observed trend. Critical cut length could be measured with $\pm 1 \text{ cm}$ uncertainty and slope angles with $\pm 2^\circ$ uncertainty. **d** Mode II energy release rate at the onset of unstable crack propagation as a fraction of the total energy release rate. The mode II fraction increases rapidly with inclination for downslope cuts ($\varphi > 0^\circ$) but only moderately for upslope cuts ($\varphi < 0^\circ$). All data points (site A, site B with extra load, literature data) follow the same trend with little scatter. The literature data set comprises almost no mode II contribution, even at inclinations as high as 36° . Mode-ratio uncertainties were calculated from error propagation of uncertainties in cut length ($\pm 1 \text{ cm}$), inclination ($\pm 2^\circ$), and weak-layer thickness ($\pm 1 \text{ mm}$). **e–g** Stacked histograms truncated at 20 counts per bin. The distribution of tested inclinations (**e**) shows that, historically, propagation saw tests were predominantly performed in flat terrain and cut upslope (orange) while the present study (green) focuses on steep downslope cuts. Critical cut lengths (**f**) are distributed uniformly with mean and standard deviation of $a_c = 31.0 \pm 14.4 \text{ cm}$ indicating equal likelihood for a wide range of cut lengths. The distribution of mode II fractions of the total energy release rate (**g**) shows that the literature data (orange) contains no information on mode II energy release rates while the present study (green) covers the full range between pure mode I and pure mode II fracture toughness of weak layers.

Remark 2.3 3) In Fracture Mechanics the concepts of fracture toughness (K_c) and fracture energy= critical energy release rate (G_c) of a material are under certain conditions equivalent but not the same. Nevertheless, sometimes G_c is referred to fracture toughness. This issue deserves at least a comment.

We agree fully and have added a comment to the manuscript. See also our answer to your remark 1) above.

Because of the confinement of the anticrack, we measure fracture toughness in terms of critical energy release rates G_c rather than stress intensity factors K_i , which take on complex values for interfacial cracks.

Remark 2.4 4) Is there any physically based motivation for proposing an interaction power law in terms of the components of the critical ERR with different exponents? I am asking because in a quite comprehensive review of closely related interaction laws for fracture energy in spring (Winkler) models by Mantic et al (Int. J. Fract., 2015, Vol 195, pages 15–38), and apparently also in the literature cited therein, only power laws with equal exponents are considered. Have you tried to perform fitting of your experimental results by considering the same exponents?

Thanks for this relevant comment. We understand the power law as a phenomenological formulation of an energetic mixed-mode interaction criterion. Because it is phenomenological, we could not formulate a physical argument for why the exponents should or should not be equal. This is echoed in studies on the fracture toughness of porous materials.¹³ Hence, we chose to not make restrictions. In the end, the best fit based on the orthogonal-distance regression was for unequal exponents. Note that the differences in the best-fit results between equal exponents or exponents determined without restrictions are minimal. For comparison:

Fit results	free exponents	forced $n = m$
n	0.20	0.27
m	0.45	0.27
\mathcal{G}_{Ic}	0.56	0.56
\mathcal{G}_{IIc}	0.79	0.79
χ_v^2	3.31	3.32

The fact that the fracture toughness values obtained from both methods are the same to within two significant digits speaks to the robustness of the method.

Many of the other interaction laws discussed by Mantič et al.¹⁴ are similar in structure and do not provide better fitting results. We have added a remark about this to the Interaction-law identification section in the supplement:

Many other interaction laws proposed in literature^{13,14} are of similar characteristics as Eq. (S8), yet these did not provide a better fit to our data.

Remark 2.5 3) Figure 4. *If some symbols related to the snow profile are included, the mechanical engineering community might not understand the meaning. Some brief comments or reference to the list of such symbols would be useful.*

We have added the following clarification and reference of the symbols in the caption of Fig. 4:

Grain types indicated in the figure legend are precipitation particles (PP), decomposing and fragmented precipitation particles (DF), rounded grains (RG), faceted crystals (FC), depth hoar (DH), surface hoar (SH), melt forms (MF), and ice formations (IF).¹⁵

Remark 2.6 4) Lines 415-423. *Please refer to Figure 1 where the specimen and test configuration is represented. More references would be helpful.*

Thank you for the comment. We have added this cross reference and have improved cross referencing across the whole manuscript. Here specifically, we have added the reference to Fig. 1.

MMFTs were then performed at different angles after 12 weights were placed on the slab (Fig. 1a,b).

Remark 2.7 5) *Mechanical model. If N is the normal force in beam model, then the parameter N in Figure 1 should be denoted by another letter or this notation issue clarified.*

Good catch. The number of experiments is now denoted $|S|$ as the cardinality of the set S of experimental samples.

Reviewer #2 (supplementary materials)

Remark 2.8 1) *General comment. Sometimes it is difficult to follow the explanations in the article and supplementary material and more cross-references to equations and sections would be very helpful. For example, in section "Data fitting procedure" the vector β is not explained, it is described later in section "Interaction-law identification" and also in the main manuscript. Some cross-references could make it clearer.*

Thanks for this comment that helped improve the clarity of the manuscript. We have improved cross referencing across the whole manuscript. Specifically here, the vector β is now explained directly after Eq. (S1). We have also added a cross reference to Eq. (1) in the main text, which provides additional context:

The interaction laws examined in this work are two-dimensional implicit nonlinear models

$$0 \approx r(\mathbf{x}_i; \beta), \quad (1)$$

where $\beta = (\mathcal{G}_{Ic}, \mathcal{G}_{IIc}, n, m)^\top$ is the vector of model parameters and $\mathbf{x}_i = (\mathcal{G}_I, \mathcal{G}_{II})_i^\top$ is the vector of independent variables, i.e., the vector of $i = 1, \dots, N$ observations (see Eq. (1) of the main text).

Remark 2.9 2) General comment. It seems that it is not mentioned in the article that there is supplementary material. A comment would be helpful (in the same way that codes availability is mentioned).

Thanks for pointing this out. We have improved cross referencing overall and now point the reader to the supplement everywhere additional relevant content can be found.

Remark 2.10 3) Section “Identification-law” verification: The objective of this section is not clear. Are you comparing r_1 and r_2 ? Are you applying both?

The objective of the section was to bring transparency into our process of selecting a suitable interaction law. For this purpose, we examined different classical formulations of interaction laws, among them r_2 . Since interaction laws of the form of r_2 do not fit well to our data, we opted not to pursue them further and chose power-law interactions of the form of r_1 instead. We now clarify this at the start of this section:

Fracture-toughness interaction laws have been proposed and examined by many authors.¹⁴ In the following, we illustrate their differences and justify our choice of a power-law interaction model using two characteristic examples.

Remark 2.11 4) Section “Finite element model”: a few details are given about the numerical model. How do you define the weak layer elements? Are you using cohesive zone model for example?

The finite-element model is a static analysis of discrete geometric configurations of a certain crack length. This suffices for the computation of displacements, stresses, and energy release rates of weak-layer anticracks of specific lengths. We do not use any crack advancement techniques such as cohesive elements, XFEM, or phase-field methods. We clarified this in the description of the finite-element model:

Weak-layer cracks are introduced by removing all weak-layer elements on the crack length a . We perform a static analysis of discrete geometric configurations with specific crack lengths a . Crack advance is not considered.

Remark 2.12 5) Section “Model Validation”: Although it is mentioned, the explanations in this section are closely related to those in References [11-13]. The authors should clarify the novelty of contributions of this part of the present work regarding the previous ones, including other apparently simpler and less general proposal by Chiaia et al. (Cold Regions Science and Technology, 2008, Vol 53, pages 170-178).

We now more clearly highlight the contributions of the present study and added references to the related proposals of McClung,¹⁶ Heierli and Zaiser,¹⁷ and Chiaia et al.¹⁸

In comparison to previous studies that used similar concepts to model the structural response of layered snowpacks,^{16–18} in our model we also consider slope normal deformations and the compliance of the weak layer. The latter effect has a substantial influence on the modeled energy release rate, which is central in this study.

Remark 2.13 6) Regarding the cited successful applications of finite fracture mechanics approaches with weak-interface models, the seminal contribution by Cornetti et al (Int J Solids Struct, 2012, 49, pages 1022-1032) also could be mentioned.

Thank you for the suggestion, we added the reference:

This has been discussed in detail by Leguillon¹⁹, laying the foundation for the successful application of finite fracture mechanics approaches with weak-interface models.^{20–22}

Reviewer #3

Remark 3.1 *The title and abstract of the submitted paper promise to talk about highly porous materials and, in particular, their fracture toughness. The abstract asserts to present new experimental tests to found (to measure? to calculate?) fracture mechanics proprieties in natural snowpack.*

Thank you for your thoughts. To clarify, yes! We measure the fracture toughness of weak snow layers, an exceptionally highly porous material found in Nature. Measuring mechanical properties of a material always needs a model to interpret the experiments. Such models can be as simple as

$$\sigma_y = \frac{F_{\text{loadcell}}}{A_{\text{specimen}}}, \quad (2)$$

for the tensile strength σ_y in uniaxial tension tests or the Irwin–Kies equation

$$\mathcal{G}_I = \frac{F}{2w} \frac{dC}{da}, \quad (3)$$

derived from beam kinematics for double-cantilever-beam (DCB) test. Because snow is very fragile, specimen geometries are limited and we need more complex models to interpret the simple test geometries and measure material properties. The model presented here is of closed form like the Irwin–Kies equation.

Remark 3.2 *Then, using a mechanical model to interpret the data (what data measured in the field? Densities?), the authors calculate the fracture toughness for anticrack growth from pure shear to pure collapse (I think I did not understand very well the use of the word “collapse” in this paper . . .).*

Thank you for your comment and for seeking clarification on the terminology used in our paper. In the context of our study, when we refer to “collapse,” we are describing the compaction of the weak layer—a localized volume loss due to compressive forces acting on the snow. This term is commonly used within the snow science community to denote scenarios where snow layers under a slab compact or settle significantly, often due to the weight of the overlying snow or external loads. This phenomenon is critical in understanding snow slab stability and is a key factor in avalanche formation.

In our experiments, we measure all properties required to accurately model snow slab behavior and fracture mechanics, as described in the Methods section and the Supplement. These include the density of all discernible slab layers, slab geometry with layer thicknesses and locations, weak-layer thickness, slope inclination, and the critical cut length—defined as the point where an artificially introduced crack becomes unstable.

Remark 3.3 *However, the paper is entirely dedicated to the presentation of new and (debatable) experimental and numerical results of a single type of snow (surface hoar). There are no comparisons/discussions of the results with other types of highly porous materials. This is both from an experimental and theoretical point of view. In fact, porosity is not a protagonist of the paper. Nor are porous materials. For this reason alone, the reader is disappointed with the content of the paper compared to the expectations provided by the title.*

We acknowledge your concerns regarding the scope of the study and the focus solely on surface hoar as the type of snow analyzed. Snow, as a natural material, presents unique challenges in terms of experimental control, particularly with the variability of weak layers that form in different winter seasons. During the period of our experiments, we could only perform experiments on surface-hoar layers, somewhat limiting the scope of our immediate study.

However, it is important to highlight that this work provides significant new insights into the properties of and fracture phenomena related to surface hoar that were previously unknown and unquantified. Moreover, while the focus was on a single type of snow, the methodologies and analytical techniques developed are indeed applicable more broadly. The study is designed as a blueprint for future experiments on other types of snow and other types of porous materials, allowing to establish a comprehensive material database that includes a variety of densities, porosities, and microstructural morphologies and their impact on fracture properties.

Your feedback on this point is highly appreciated, and we have made revisions to the manuscript accordingly. To enhance the manuscript, we have included a discussion on the fracture analysis of other porous materials, providing readers with a broader understanding of the topic. Additionally, in revising the rest of the document, we have aimed to clearly articulate the generality of our experimental approach and the model-driven evaluation of the experimental results, as well as the comprehensive nature of our experimental dataset.

Other porous materials are often described using similar power-law type interaction laws, with either equal or unequal exponents to capture the relationships between stress intensity factors under mixed loading conditions.^{13,23} Common predictors for fracture toughness across highly porous materials include density²⁴ and microstructure,^{25,26} where higher density generally correlates with increased fracture toughness.²⁷ Despite extensive literature on the tensile and bending properties of porous materials, studies focusing on compressive fracture properties remain scarce, highlighting a pervasive challenge in understanding fracture behaviors under compressive loads.^{13,26} The concept of anticracks under compressive mode I loading has been explored in man-made materials like glass foams²⁸ and 3D printed brittle open-cell structures,²⁹ and studies indicate that the morphology of anticracks under compression resembles tensile mode I cracks.²⁸ Nevertheless, there is a notable absence of experiments measuring mixed compression and shear fracture toughness across various materials, especially natural porous materials such as snow.

Remark 3.4 *The authors write an introduction (overview) briefly summarising the bibliography on porous media, fracture mechanics, and then give a lot of space to fracture mechanics and related experimental tests on snow material. It is not clear why this work is also mentioned: Kiakojouri, F., De Biagi, V., Chiaia, B. & Sheidaii, M. R. Progressive collapse of framed building structures: Current knowledge and future prospects. Engineering Structures 206, 110061 (2020). URL <https://www.sciencedirect.com/science/article/pii/S0141029619322576>. It is true that the authors state that the mechanism of snowpack collapse is progressive, but the cited paper: 1) is a review of the state-of-art on progressive collapse on frame structures (reinforced concrete, steel,...); 2) it is a comparison of the various robustness techniques of framed building structures; 3) it discusses pure and mixed progressive collapse mechanisms, therefore very far from the subject of the submitted paper.*

With this citation, we wanted to show that collapse phenomena occur in many different settings. However, given your justified concerns, we removed the reference.

Remark 3.5 *The overview mentions the experimental PST tests (used to measure of fracture properties in snow, exclusively) and it proposes MMFT test for, I think, the measurement of fracture toughness of the (snow) weak-layer in mixed mode. The MMFT experimental test is presented here for the first time. It is my opinion that authors must be very careful with respect to the parameter that they want and think to measure (which may not coincide). Surely, the paper is the result of serious and remarkable work: 1) The effort to design and implement a new experimental test to understand the of slab avalanche release; 2) the development of a numerical model; 3) the extensive experimental campaign. In my opinion, the work will not be significant for the field and related fields. The theory used (the model) is already presented and known (also) in the snow and, unfortunately, the data (and not the parameters calculated with the numerical model) from the extensive experimental campaign (the real novelty of the article) may not be used to validate other models as they are flawed by the imposed overload.*

Thank you for your constructive feedback and positive assessment of our work. We appreciate the opportunity to clarify a few key aspects mentioned in your review:

Our model is fundamentally analytical, not numerical. The implementation we have adopted includes two distinct numerical computation steps: (i) solving the eigenvalue problem that arises from the fundamental solution to the governing system of differential equations, and (ii) solving the linear system of equations derived from the boundary and transmission conditions. Here, "numerical" pertains to specific computation steps—inserting values into an established system—rather than approximate methods such as finite difference or finite element techniques. This choice primarily enhances implementation efficiency. As detailed in the supplementary materials,

it is possible to express the entire equation set symbolically. We have confirmed that performing symbolic computations—inserting values only in the final step—produces results identical to those obtained through our "numerical" approach, where values are input prior to solving the eigenvalue problem.

The additional applied load is not a limitation of our methodology; rather, it is crucial for achieving the necessary number and precision of experiments. For example, conducting tests on full-height slabs would necessitate such a substantial volume of snow—owing to the required length of propagation saw tests (PSTs)—that it would be impractical to gather adequate samples on the same weak layer.

To utilize our dataset for validating other models, we recommend replicating the complete geometry, including surface loads, and predicting the measured crack lengths. Our dataset is among the most comprehensive available for this purpose, providing detailed information to facilitate accurate model validation. Moreover, the measured fracture toughness provides a unique and crucial resource for calibrating and validating numerical models of crack propagation, such as material-point or discrete-element models.

Remark 3.6 *The paper does not present any noteworthy novelties compared to the established literature, (among which the following are missing: Ritter, J., Löwe, H. & Gaume, J. Microstructural controls of anti-crack nucleation in highly porous brittle solids. *Sci Rep* 10, 12383 (2020). <https://doi.org/10.1038/s41598-020-67926-2> Heierli, J., Gumbsch, P. & Zaiser, M. Anticrack nucleation as triggering mechanism for snow slab avalanches. *Science* 321, 240–243. <https://doi.org/10.1126/science.1153948> (2008). Mulak, D. & Gaume, J. Numerical investigation of the mixed-mode failure of snow. *Comput. Part. Mech.* 6, 439–447. <https://doi.org/10.1007/s40571-019-00224-5> (2019)), but only a different (numerical) analysis of the mixed mode fracture toughness of the weak-layer proposed in the form of power-law (Eq. 1), mimicking classical fracture mechanics.*

We believe that our research introduces substantial novelty in the field. Please allow us the opportunity to convince you of this. To the best of our knowledge, there are no existing studies that measure fracture toughness across the complete spectrum of mode interactions for weak snow layers or any similarly highly porous media that could fail in a shear-compression mixed mode.

While PSTs and other fracture mechanical tests, combined with mechanical or numerical models, have been utilized to determine fracture toughness in porous media including snow, these studies typically focused on measuring pure mode II or mode I fracture toughness in laboratory settings,^{30,31} or reported mixed-mode fracture toughness from field tests without specific details on the mode loading.^{1,2} Only recently has the reporting of mixed-mode fracture toughness, including the contributions of different modes, begun.³² Our current work demonstrates that mixed-mode fracture toughness values reported in literature were governed by compression and shear fracture toughness values have rarely been measured (Fig. 2). Previously, a methodology to systematically evaluate all mixed-mode fracture toughness compositions—from shear to compression—for the same weak layer was unavailable.

Our study distinctly sets itself apart from the literature cited above by employing a field-applicable test that allows examination of the same weak layer across all mixed-mode compositions. We utilize a closed-form analytical solution to model the experiments and directly determine fracture toughness from field experiments. In contrast, numerical methods like the mentioned discrete element modeling (DEM) often involve a somewhat circular reasoning—assuming specific particle interaction laws (e.g., stress-based micro-mechanical failure mechanisms, friction) to simulate the macroscopic material response. A fact correctly criticized by the reviewer, but not applied in the present work. Indeed, while model input parameters are typically validated against measurements like experimentally determined failure envelopes, validation against a fracture envelope was previously unfeasible due to its absence in literature. Our work now provides the necessary data to enable such validation, thereby significantly enhancing the robustness and applicability of numerical models.

Remark 3.7 *The work supports its claims, but these are as trivial as “Cut lengths increase with slope angle”.*

While it may appear that the observation "cut lengths increase with slope angle" is trivial, for the first time, this conclusion is supported by a robust set of data and represents a significant clarification in a previously ambiguous field of study. Prior research on this topic has shown conflicting results, with studies indicating varying trends—some suggested cut lengths decrease with slope angle,³³ others reported an increase,^{34,35} and still others found no discernible trend at all.³⁶

Our work aims to provide clarity and resolve these discrepancies by offering comprehensive experimental evidence that supports a consistent trend. It is well-known that avalanches are typically triggered in steep terrain, suggesting that weak-layer anticracks might propagate more readily on steeper slopes. This would intuitively correspond to shorter critical cut lengths at steeper angles. However, our results indicate otherwise, challenging prevailing assumptions and contributing new insights into the mechanics of snow slab release. The findings are an essential contribution to the field of snow science, enhancing our understanding of avalanche formation related to slope angles and informing safety protocols and preventive measures.

Remark 3.8 *With regard to experimental tests and subsequent analysis/interpretation, I do not understand the need to carry out an experimental campaign with a new test (with even additional loads) when unconventional snow fracture tests (PST) have recently been validated by the scientific community.*

While the use of traditional PSTs may initially appear attractive, our research identified several limitations inherent in the classical PST setup that preclude the collection of our comprehensive dataset. In classical setups, the thicker the slab, the longer the required PST beam. These setups typically engage the full slab depth, which ranges from 0.5 to over 1 meter. According to the literature, to mitigate edge effects from the free beam ends, beam lengths must be at least 3–4 times the slab thickness.³⁷ However, for steep slope angles, this guideline proves insufficient, as edge effects vary significantly with the slope angle (see this preprint <https://doi.org/10.5194/egusphere-2024-690>). An unaltered PST design would necessitate beams several meters long, resulting in experiments that weigh hundreds of kilograms and are both time-consuming and spatially demanding. Additionally, such sizes make it impractical to excavate and tilt the samples.

To address these challenges, we adapted the method by substituting a significant portion of the slab with a dead load. This adjustment reduces slab thickness and stiffness, and reintroduces weight onto the slab surface, effectively circumventing the original method's limitations. Moreover, upslope cuts are traditionally made in PSTs, which inherently limits the mode II contribution due to the slope inclination, as demonstrated in the present study.

In conclusion, classical PSTs predominantly measure compression-driven fracture toughness, as illustrated in Figure 1g. Our modifications are crucial to accurately measure the fracture toughness of the same weak layer across the entire spectrum, extending to pure shear.

Remark 3.9 *Furthermore, I do not understand the reason for the imposed overload and its way to apply (why a distributed load and not a point load?). For the spontaneous release of snow avalanches, the 'load' of the slab above the weak layer would have been sufficient . . .*

We agree with the reviewer, that other forms of additional load or the mere gravitational loading can be used in propagation saw test. We use the additional load to control the critical cut length, which is important to obtain a high mode II ratio. Using a point load, as suggested by the reviewer, would also work in the same way, but we found it experimentally more easy to use a distributed load. To make this more clear, we have improved the respective section in the manuscript:

By adding variable surface dead loads in the form of steel bars, we can then control the cut length a priori to increase the mode II contribution.

Remark 3.10 *Instead of the mixed-mode interaction law for weak-layer anticracks, it would have been more interesting to have the influence of gravity (and inclination) with respect to the in-plane GI and GII or directly a formula (like the classical fracture mechanics formulas) that gives the G (or K) dependent on the specimen size, crack length and/or other measurable quantities.*

Indeed, exploring the influence of gravity and slope inclination on the energy release rates \mathcal{G}_I and \mathcal{G}_{II} would offer a valuable perspective on fracture mechanics in this context. Although not the focus of this study, it provides the tools necessary to perform this analysis. The implementation provided alongside the manuscript takes inputs like geometry and material properties and outputs energy release rates (and if needed, field quantities such as displacements and stresses). It functions in the same way as one-line equations for \mathcal{G} or K . As noted above, the complex layered nature of the slab and the compliance of the weak layer introduce complexities that, if modeled accurately, preclude straightforward expressions of one-line equations for \mathcal{G} or K .

Additionally, now that fundamental material parameters (fracture toughness) have been determined through our study, the impact of gravity and inclination on physical systems can be analyzed using any suitable model, whether it is analytical or numerical. This flexibility enables a broader application and validation of the results across different modeling frameworks, enhancing the practical utility and theoretical understanding of fracture mechanics in these complex highly porous systems.

Remark 3.11 *Although 'energetic', the methodology is based on the theory of laminated plates under cylindrical bending with the weak-layer modelled as an elastic (Winkler) foundation, a typically tensional model (into which stiffnesses enter). From this, using a simple Mohr-Coulomb resistance criterion, the authors calculate the normal and tangential stresses (at the crack tip) and, consequently, the respective $G = GI + GII$. Therefore, G or GI or GII are not calculated directly as for standard tests (e.g., 3PBT), but calculated by means of a numerical tension model*

As discussed above, every mechanical test requires a certain model for its interpretation and for the derivation of material properties. In a notched three-point bending test, this could be, for instance,

$$K_I = \frac{4P}{B} \sqrt{\frac{\pi}{W}} \left[1.6 \left(\frac{a}{W} \right)^{1/2} - 2.6 \left(\frac{a}{W} \right)^{3/2} + 12.3 \left(\frac{a}{W} \right)^{5/2} - 21.2 \left(\frac{a}{W} \right)^{7/2} + 21.8 \left(\frac{a}{W} \right)^{9/2} \right] \quad (4)$$

according to Bower (Applied mechanics of solids. CRC Press, 2009) or

$$K_I = \frac{6P}{BW} \sqrt{a} \frac{1.99 - a/W(1 - a/W) (2.15 - 3.93a/W + 2.7(a/W)^2)}{(1 + 2a/W)(1 - a/W)^{3/2}} \quad (5)$$

according to ASTM D5045-14 and E1290-08. Both models represent approximations to the true value of K_I .

We take the same approach of deriving a set of equations that allow us to calculate a quantity of interest from our experiments. Our model is closed form, i.e., does not require numerical methods and we do not make use of the Mohr-Coulomb criterion in deriving fracture toughness. Because of the layered nature of snowpacks and because of the asymmetry of the experimental setup, the derived equations are longer than those of the classical DCB or 3PB tests. For this reason, we offer a Python implementation of the model for download and provide the details of the derived equations in the supplementary methods.

Remark 3.12 *For those of us in classical fracture mechanics, it is a big difference between G 'energy release rate' and K (or K_c) 'fracture toughness', which are linked together by the modulus of elasticity. I would ask the authors to do a thorough check to see if the fracture toughness they indicate is not instead an energy release rate. I would ask the authors to report the quantities with their symbol from classical fracture mechanics.*

Thank you for the remark. This deserves some attention to detail. It is our understanding that both \mathcal{G}_c and K_c are referred to as fracture toughness in text books on fracture mechanics.^{9,38,39} Of course, one is the critical energy release rate and the other is the critical stress intensity factor. As in the present case, the anticrack is confined within the weak layer, the singularity exponent characterizing the crack-tip near field, whose strength is expressed by the stress intensity factor K , can take complex values. For this reason, we measure fracture toughness in terms of critical energy release rates and denote them as \mathcal{G}_{Ic} and \mathcal{G}_{IIc} as in classical fracture mechanics.^{9,38,39}

Please also see our response to **Remark 2.1** of **Reviewer #2** who raised a similar concern. We added the following clarification to the manuscript:

To enhance our understanding of the fracture behavior of porous materials under mixed-mode loading involving both closing mode I and mode II, we introduce a modification of the conventional fracture mechanical experiment PST. This methodology provides insight into previously unexplored fracture regimes. The design has the advantage that the anticrack is confined to the weak layer. As a result, the mode mixity of the crack tip loading remains fairly constant during crack growth because the anticrack cannot kink into pure mode I loading conditions. A similar geometry was used for some of the first measurements of supersonic shear cracks,⁸ a phenomenon recently observed for anticracks confined in weak snow layers.^{10,11} Because of the confinement of the anticrack, we measure fracture toughness in terms of critical energy release rates \mathcal{G}_c rather than stress intensity factors K_I , which take on complex values for interfacial cracks.

Remark 3.13 *The methods are detailed enough to allow the work to be reproduced. The method is “circular”: the authors tries to validate a numerical model that attempts to reproduce an experiment in the field with the aim is to determine an unmeasurable physical quantity. In my opinion, the process is too tweakable, risking that the physical quantities involved in the model lose their own physical meaning.*

Thank you for your comment and critique regarding circular modeling frameworks. However, we respectfully disagree as our method is neither numerical nor circular. We use experimentally measured quantities, such as snow densities, inclination angle and critical crack length, to derive fracture toughness values. This ensures our results are empirically grounded and independently verifiable. Our stated goal is to derive toughness values from field experiments, not to validate our model.

Our methodology is akin to the characterization of mode I tensile fracture toughness using a three-point bending test as per ASTM D5045-14 standards. The primary difference in our approach lies in the geometry considered and the resulting complexity of the analytical equations, which are tailored to the unique challenges presented by snow mechanics. This adherence to empirical measurement ensures that our findings retain their physical significance and provide a robust basis for further research and model validation within the snow science community.

Note that we share your concerns about circular reasoning associated with models where an inter-particle interaction law is defined—such as in finite-element, discrete-element, or material-point models—and then used to predict macroscopic material behaviors. Indeed, in these frameworks, the micro-mechanical laws typically define properties like elasticity and strength, which can inadvertently predetermine other properties such as fracture toughness. This leads to a situation where the derived macroscopic properties are essentially artifacts of the inputs rather than independent measurements, thus risking the physical meaningfulness of these quantities. Our work, on the other hand, explicitly addresses and mitigates this issue. By providing direct measurements of mixed-mode fracture toughness, our research serves as a crucial resource for calibrating these micro-mechanical models more accurately, thereby reducing the methodological circularity often seen in such approaches.

We greatly appreciate the feedback provided by the reviewers, which has significantly enhanced the quality and clarity of our manuscript. We have carefully considered each comment and made corresponding revisions to address the points raised. We believe these changes have strengthened our arguments and findings, and we hope that the manuscript now meets the high standards of Nature Communications. We look forward to the possibility of our work contributing to the field. Thank you for your thoughtful consideration and guidance throughout this revision process.

References

- [1] Bergfeld, B. *et al.* Dynamic crack propagation in weak snowpack layers: insights from high-resolution, high-speed photography. *The Cryosphere* **15**, 3539–3553 (2021).

- [2] van Herwijnen, A. et al. Estimating the effective elastic modulus and specific fracture energy of snowpack layers from field experiments. *Journal of Glaciology* **62**, 997–1007 (2016).
- [3] Birkeland, K. W., van Herwijnen, A., Reuter, B. & Bergfeld, B. Temporal changes in the mechanical properties of snow related to crack propagation after loading. *Cold Regions Science and Technology* **159**, 142–152 (2019).
- [4] Broberg, K. B. The near-tip field at high crack velocities. *International Journal of Fracture* **39**, 1–13 (1989).
- [5] Ritter, J., Löwe, H. & Gaume, J. Microstructural controls of anticrack nucleation in highly porous brittle solids. *Scientific Reports* **10**, 12383 (2020).
- [6] Shaikheea, A. J. D., Cui, H., O'Masta, M., Zheng, X. R. & Deshpande, V. S. The toughness of mechanical metamaterials. *Nature Materials* **21**, 297–304 (2022).
- [7] Hedvard, M. L., Dias, M. A. & Budzik, M. K. Toughening mechanisms and damage propagation in architected-interfaces. *International Journal of Solids and Structures* **288**, 112600 (2024).
- [8] Rosakis, A. J., Samudrala, O. & Coker, D. Cracks Faster than the Shear Wave Speed. *Science* **284**, 1337–1340 (1999). URL <https://www.science.org/doi/10.1126/science.284.5418.1337>.
- [9] Freund, L. B. *Dynamic Fracture Mechanics* (Cambridge University Press, Cambridge, 1990).
- [10] Trottet, B. et al. Transition from sub-Rayleigh anticrack to supershear crack propagation in snow avalanches. *Nature Physics* **18**, 1094–1098 (2022).
- [11] Siron, M., Trottet, B. & Gaume, J. A theoretical framework for dynamic anticrack and supershear propagation in snow slab avalanches. *Journal of the Mechanics and Physics of Solids* **181**, 105428 (2023).
- [12] Bobillier, G. et al. Numerical investigation of crack propagation regimes in snow fracture experiments. *Granular Matter* **26**, 58 (2024).
- [13] Maršavina, L. & Linul, E. Fracture toughness of rigid polymeric foams: A review. *Fatigue & Fracture of Engineering Materials & Structures* **43**, 2483–2514 (2020).
- [14] Mantič, V., Távara, L., Blázquez, A., Graciani, E. & París, F. A linear elastic-brittle interface model: application for the onset and propagation of a fibre-matrix interface crack under biaxial transverse loads. *International Journal of Fracture* **195**, 15–38 (2015). URL <http://link.springer.com/10.1007/s10704-015-0043-0>.
- [15] Fierz, C. et al. The international classification for seasonal snow on the ground. *IHP-VII Technical Documents in Hydrology* **83** (2009).
- [16] McClung, D. M. Shear Fracture Precipitated by Strain Softening as a Mechanism of Dry Slab Avalanche Release. *Journal of Geophysical Research* **84**, 3519–3526 (1979).
- [17] Heierli, J. & Zaiser, M. An analytical model for fracture nucleation in collapsible stratifications. *Geophysical Research Letters* **33**, L06501 (2006).
- [18] Chiaia, B. M., Cornetti, P. & Frigo, B. Triggering of dry snow slab avalanches: stress versus fracture mechanical approach. *Cold Regions Science and Technology* **53**, 170–178 (2008).
- [19] Leguillon, D. Strength or toughness? A criterion for crack onset at a notch. *European Journal of Mechanics – A/Solids* **21**, 61–72 (2002).
- [20] Cornetti, P., Mantič, V. & Carpinteri, A. Finite Fracture Mechanics at elastic interfaces. *International Journal of Solids and Structures* **49**, 1022–1032 (2012).
- [21] Weißgraeber, P. & Becker, W. Finite Fracture Mechanics model for mixed mode fracture in adhesive joints. *International Journal of Solids and Structures* **50**, 2383–2394 (2013).
- [22] Rosendahl, P. L., Staudt, Y., Schneider, A. P., Schneider, J. & Becker, W. Nonlinear elastic finite fracture mechanics: Modeling mixed-mode crack nucleation in structural glazing silicone sealants. *Materials & Design* **182**, 108057 (2019).
- [23] Marsavina, L. et al. Refinements on fracture toughness of PUR foams. *Engineering Fracture Mechanics* **129**, 54–66 (2014).
- [24] Linul, E., Maršavina, L., Vălean, C. & Bănică, R. Static and dynamic mode I fracture toughness of rigid PUR foams under room and cryogenic temperatures. *Engineering Fracture Mechanics* **225**, 106274 (2020).
- [25] Arakere, N. K., Knudsen, E. C., Wells, D., McGill, P. & Swanson, G. R. Determination of mixed-mode stress intensity factors, fracture toughness, and crack turning angle for anisotropic foam material. *International Journal of Solids and Structures* **45**, 4936–4951 (2008).
- [26] Khosravani, M. R., Berto, F., Ayatollahi, M. R. & Reinicke, T. Fracture behavior of additively manufactured components: A review. *Theoretical and Applied Fracture Mechanics* **109**, 102763 (2020).

- [27] Poapongsakorn, P. & Carlsson, L. A. Fracture toughness of closed-cell PVC foam: Effects of loading configuration and cell size. *Composite Structures* **102**, 1–8 (2013).
- [28] Heierli, J., Gumbsch, P. & Sherman, D. Anticrack-type fracture in brittle foam under compressive stress. *Scripta Materialia* **67**, 96–99 (2012).
- [29] Shenhav, L. & Sherman, D. Fracture of 3D printed brittle open-cell structures under compression. *Materials & Design* **182**, 108101 (2019).
- [30] Sigrist, C., Schweizer, J., Schindler, H.-J. & Dual, J. The energy release rate of mode II fractures in layered snow samples. *International Journal of Fracture* **139**, 461–475 (2006). URL <http://link.springer.com/10.1007/s10704-006-6580-9>.
- [31] Sigrist, C. & Schweizer, J. Critical energy release rates of weak snowpack layers determined in field experiments. *Geophysical Research Letters* **34** (2007).
- [32] Bergfeld, B. *et al.* Temporal evolution of crack propagation characteristics in a weak snowpack layer: conditions of crack arrest and sustained propagation. *Natural Hazards and Earth System Sciences* **23**, 293–315 (2023).
- [33] Gaume, J., van Herwijnen, A., Chambon, G., Wever, N. & Schweizer, J. Snow fracture in relation to slab avalanche release: critical state for the onset of crack propagation. *The Cryosphere* **11**, 217–228 (2017).
- [34] Gauthier, D. & Jamieson, B. Evaluation of a prototype field test for fracture and failure propagation propensity in weak snowpack layers. *Cold Regions Science and Technology* **51**, 87–97 (2008). URL <http://linkinghub.elsevier.com/retrieve/pii/S0165232X07000821>.
- [35] McClung, D. M. Dry snow slab quasi-brittle fracture initiation and verification from field tests. *Journal of Geophysical Research* **114**, F01022 (2009).
- [36] Bair, E. H., Simenhois, R., Birkeland, K. & Dozier, J. A field study on failure of storm snow slab avalanches. *Cold Regions Science and Technology* **79–80**, 20–28 (2012). URL <https://www.sciencedirect.com/science/article/pii/S0165232X12000547>.
- [37] Bair, E. H., Simenhois, R., Van Herwijnen, A. & Birkeland, K. The influence of edge effects on crack propagation in snow stability tests. *The Cryosphere* **8**, 1407–1418 (2014).
- [38] Broberg, K. B. *Cracks and fracture* (Elsevier, 1999).
- [39] Anderson, T. L. *Fracture Mechanics* (CRC Press, Boca Raton, 2017), 4th edn.

REVIEWERS' COMMENTS

Reviewer #1 (Remarks to the Author):

The authors have sufficiently answered all of my questions. The additions of Figure 5 and the accompanying discussion were particularly helpful.

Reviewer #2 (Remarks to the Author):

The authors have properly answered all the questions raised by the reviewer and have made appropriate changes to the manuscript. The revised manuscript is therefore recommended for publication.

Minor correction: In the supplementary material, the sentence below Eq. (S1) "the vector of $i = 1, \dots, N$ observations" should be changed to the vector of $i = 1, \dots, |S|$ observations, I think.

Reviewer #3 (Remarks to the Author):

Thanks to the authors for the clarifications and corrections.

I really appreciated the review of the article and of the supplementary information

The paper is more comprehensible with the inclusion of new paragraphs (e.g., Asymmetry in energy release rates) and more detailed figure captions. I suggest the authors consider whether to keep such extensive and lengthy captions or to insert the details of figures in the text and shorten the caption.

I really appreciated the revision of the paragraphs 'Results - Weak-layer anticracks are controlled by fracture toughness' and 'Discussion - Implications and limitations for avalanche modelling'.

Thanks to this revision, the discussion in the field of fracture mechanics is much clearer.

Typing errors:

Page 5 - left column - Line 8 (approx.) add a space between 'I' and 'loading'.

Page 5 - left column - Line 12 (approx.) add a space between 'I' and 'cracks'.

Page 6 - right column - Line 8 (approx.) of paragraphs "Asymmetry in energy release rates", add a space between 'II' and 'contributions '.

Page 7 - left column - Line 19 (approx.) add a space between "II" and "contributions".

Page 7 - left column - Line 23 (approx.) add a space between "II" and "ratio".

Page 7 - left column - Line 24 (approx.) add a space between "II" and "condition".

Page 7 - left column - Line 26 (approx.) add a space between "I" and "anticrack

Page 7 - left column - Line 34 (approx.) add a space between ""II" and "ERR".

Beware that in the References there is still the citation:

[10] Kiakojouri, F., De Biagi, V., Chiaia, B. & Sheidaii, M. R. Progressive collapse of framed building structures: Current knowledge and future prospects. *Engineering Structures* 206, 110061 (2020).

URL <https://www.sciencedirect.com/science/article/pii/S0141029619322576>

to be removed.

I would like to thank the authors for carefully checking that the toughness of the fracture is not instead a rate of energy release, and for reporting the quantities with their symbol from classical fracture mechanics.

Surely, the paper is the result of serious and remarkable work:

- 1) The effort to design and implement a new experimental test to understand the of slab avalanche release;
- 2) The improvement of analytical models already presented,
- 3) The extensive experimental campaign.

It is my opinion that the real novelty of the paper is the extensive experimental campaign, while the theory presented is already presented and known in the snow field. In fact, the reviewed paper includes the references of well-known analytical models (McClung, Heierli and Zaiser, and Chiaia et al.) in the "Model Validation" section. In respect of which, as stated by the authors, this paper considers also the deformation field (an extension that also the mentioned models could present ...). As we well know, deformation field is nearly impossible to measure in the snow. In fact, displacements are very complicated to obtain experimentally in the snow field ('impossible' except by analysing video images). Were the displacements measured directly during the MMFT tests? The authors' response on the use of numerical methods did not convince me. I do not question the use of numerical computation steps (as stated in the response to Remark 3.5) or modified FEA (response to Remark 2.11) for model validation. But, the aim of the paper is to calculate a physical quantity (fracture toughness) that cannot be directly measured in the field using a (new) experiment methodology (yet to be scientifically validated).

This, thanks to a model validated on the results comparison with a numerical model (FEA) of Benchmark (reference) profiles. Moreover, almost all the physical quantities involved in calculating fracture toughness are not measured, but are obtained using equations from other models. In my opinion, this process is too random and tweakable: the physical quantities involved may lose their own physical meaning.

The methods are detailed enough to allow the work to be reproduced.

The sentence inserted in the "Supplementary information" following Remark 2.4 "Many other interaction laws proposed in literature 13,14 are of similar characteristics as Eq. (S8), yet these did not provide a better fit to our data." is worrying and cryptic. Why does this happen? Why cannot previous, established models replicate the data from your experiment? What do you think is the reason?

Finally, I think that the authors have not responded to my Remark 3.10. In my opinion, the "gravity" factor with respect to the MMFT test plays a major role (as the inclination).

Reviewer #4 (Remarks to the Author):

Second revision of NCOMMS-24-10218

Fracture toughness of mixed-mode anticracks in highly porous materials

Dear Editors and Reviewers,

Thank you for your thorough review and constructive feedback on our manuscript and thank you for approving of the changes to our manuscript in principle. We appreciate the opportunity add further perspectives on additional comments made to enhance the clarity, accuracy, and impact of our manuscript.

In this document, we again address each remark (*blue, italic*) point by point (black) and detail the changes we have made to the manuscript (*italic quotes*). We have attached a revised manuscript file with changes highlighted. We hope that these modifications adequately address the concerns raised and improve the manuscript. We are grateful for the guidance that the reviewers' expertise has provided and are confident that these changes have strengthened our submission.

Reviewer #3

Remark 3.14 Surely, the paper is the result of serious and remarkable work: 1) The effort to design and implement a new experimental test to understand the of slab avalanche release; 2) The improvement of analytical models already presented, 3) The extensive experimental campaign. It is my opinion that the real novelty of the paper is the extensive experimental campaign, while the theory presented is already presented and known in the snow field. In fact, the reviewed paper includes the references of well-known analytical models (McClung, Heierli and Zaiser, and Chiaia et al.) in the "Model Validation" section. In respect of which, as stated by the authors, this paper considers also the deformation field (an extension that also the mentioned models could present . . .). As we well know, deformation field is nearly impossible to measure in the snow. In fact, displacements are very complicated to obtain experimentally in the snow field ('impossible' except by analysing video images). Were the displacements measured directly during the MMFT tests?

We have not used full-field displacement measurements in this study. Nonetheless, we acknowledge the concerns and point out in our discussion:

Furthermore, since the size and shape of the interaction law are in part influenced by the choice of elastic-modulus parametrization, future experiments should be recorded on video to independently estimate the elastic properties from measured displacement fields.

Remark 3.15 The authors' response on the use of numerical methods did not convince me. I do not question the use of numerical computation steps (as stated in the response to Remark 3.5) or modified FEA (response to Remark 2.11) for model validation. But, the aim of the paper is to calculate a physical quantity (fracture toughness) that cannot be directly measured in the field using a (new) experiment methodology (yet to be scientifically validated). This, thanks to a model validated on the results comparison with a numerical model (FEA) of Benchmark (reference) profiles. Moreover, almost all the physical quantities involved in calculating fracture toughness are not measured, but are obtained using equations from other models. In

my opinion, this process is too random and tweakable: the physical quantities involved may lose their own physical meaning.

We share the concerns on uncertainties of involved parameters. For this purpose, we have supplied the sensitivity study shown in Fig. 6. To better reflect that fracture toughness values given in this work are not measured directly, but are the result of a model chain (that includes uncertainties), we have adjusted the language in the manuscript and have now softened the verbiage in several places. Nonetheless, allow us to emphasize that any fracture toughness evaluation of any laboratory experiment requires an underlying model.

Remark 3.16 *The methods are detailed enough to allow the work to be reproduced. The sentence inserted in the "Supplementary information" following Remark 2.4 "Many other interaction laws proposed in literature 13,14 are of similar characteristics as Eq. (S8), yet these did not provide a better fit to our data." is worrying and cryptic. Why does this happen? Why cannot previous, established models replicate the data from your experiment? What do you think is the reason?*

Thank you for the remark. Former works that have looked at interaction laws were concerned with isotropic materials or fiber reinforced plastics under tension and shear. In their studies, both material class and load situation were different ones. To our knowledge, no other study has attempted to model porous materials subjected to compressive failure. We have clarified this in the manuscript:

Many other interaction laws proposed in literature^{9,12}—for dense isotropic materials or fiber-reinforced plastics under combined tension and shear—are of similar characteristics as Eq. (S8). However, as they are aimed at both other classes of materials and other load interactions, they did not provide a better fit to our data.

Remark 3.17 *Finally, I think that the authors have not responded to my Remark 3.10. In my opinion, the "gravity" factor with respect to the MMFT test plays a major role (as the inclination).*

We agree that gravity and inclination are major drivers of the mode composition and of the total energy release rate in this test setup. In fact, we specifically use their effect to produce changes in the mode-II contribution to the total energy release rate. In practice, the effects of inclination and crack length cannot be separated. This is because changes in inclination cause changes in the total energy release rate, which in return, changes the critical cut length. In pure parametric studies, the model allows for independently showing both effects. For instance, we have isolated and shown in the influence of inclination on the ERR in Figure 5 in Adam, V. et al. *A field test for the mixed-mode fracture toughness of weak layers* (<http://arc.lib.montana.edu/snow-science/item/3042>). We have added the reference to the results section but believe that this specific exercise is outside the scope of the present work.

We greatly appreciate the feedback provided by the reviewers, which has again enhanced the quality and clarity of our manuscript. We have carefully considered each comment and made corresponding revisions to address the points raised. We believe these changes have strengthened our arguments and findings, and we hope that the manuscript now meets the high standards of Nature Communications. We look forward to the possibility of our work contributing to the field. Thank you for your thoughtful consideration and guidance throughout this revision process.